# LET LARGE LANGUAGE MODELS FIND THE DATA TO TRAIN THEMSELVES

## ABSTRACT

The current iterative development process for large language models (LLMs) is heavily data-centric, relying on human researchers and engineers to manually analyze model performance and determine what data to acquire for further training. However, this human-supervised approach is costly and may fail to identify optimal training signals. Its scalability is further limited as models become increasingly capable and may eventually exceed human intelligence. To address these issues, we propose an automated framework that enables models to autonomously discover and strategically acquire the most valuable training data to enhance their performance. It establishes a self-improving framework where models can invoke APIs to crawl and/or generate tailored datasets from various resources and environments, and retrain themselves. The data selection decisions are shaped by reinforcement feedback signals that reward performance gains while penalizing computational overhead. This formulation incentivizes models to develop self-knowledge about their strengths and areas for improvement in order to efficiently select training data. Empirical results demonstrate that LLMs operating within our framework are able to autonomously and strategically acquire valuable training data to enhance their performance across a variety of skills in 1,000 diverse in-house test tasks and three public benchmarks.

## 1 INTRODUCTION

Large language models (LLMs) (Zhao et al., 2023) have seen remarkable progress in various domains and tasks with extensive training on massive data. Currently, one of the most important aspects of developing LLMs is *data engineering*, which typically involves teams of researchers and engineers meticulously examining model outputs, identifying shortcomings, and then collecting and curating next-iteration training data from various sources to refine model performance (Achiam et al., 2023; Anil et al., 2023; Dubey et al., 2024). While this method has yielded impressive results, it is not without inherent limitations. The human-driven nature of this process introduces potential biases and inefficiencies, as it relies heavily on the intuition and expertise of the development team (Wang et al., 2023; Sun et al., 2024). Furthermore, as these AI models continue to evolve in complexity and capability, there is a growing concern that they may soon exceed human intelligence in certain domains, which raises questions about the long-term viability and scalability of current development practices (Burns et al., 2024).

In this work, we conceptualize and prototype an ACTIVE DATA SEARCH (ADS) framework that facilitates LLMs to automatically acquire training data from external environments for training themselves, without human supervision. In ADS, existing data collection and curation methods are encapsulated into APIs and the LLMs themselves determine when and how they use these APIs for data acquisition. The APIs may involve a wide range of data crawling, filtering, cleaning, refinement, synthesis, and manual annotation techniques to obtain tailored training data. The data sources are also varied, from raw texts extracted via web searches to documents sourced across various platforms, as well as supervised demonstrations generated by AI assistants.

We achieve the above automatic process through development of an optimizer that generates these APIs sequentially based on the target task. Specifically, the optimizer first analyzes the required capabilities for successful task completion, then strategically invokes appropriate API calls to collect training data for performance improvement. To facilitate the optimizer's decision-making process in obtaining optimal training data, we propose a reinforcement learning strategy for optimizer training,

leveraging feedback reward signals from the policy to maximize task performance while minimizing computational costs by iterative rejection sampling (RS) (Bai et al., 2022a) and direct preference optimization (DPO) (Rafailov et al., 2024). This refinement process fosters the development of self-knowledge regarding the model's strengths and weaknesses, enabling more efficient training data discovery and utilization. Compared to recent studies (Lozhkov et al., 2024; Wettig et al., 2024; Zhou et al., 2024a) that focus on improving the quality of training data agnostically, a unique characteristic of our framework is its model- and task-specific nature for data acquisition, enabling more tailored and targeted performance enhancement.

In our experiments, we build on top of two popular open-source LLMs: Qwen-2-7B-Instruct (Yang et al., 2024a) and Gemma-2-9B-Instruct (Riviere et al., 2024), and establish three distinct types of APIs for knowledge *acquisitions*, *utilization*, and *enhancement*. Experimental results across 1,000 in-house test tasks with both reward model (RM) and GPT-4 evaluation demonstrate that the LLM equipped with ADS presents consistent performance improvements, achieving a win rate of more than 80% in RM judgment compared to the initial baseline. Moreover, in three public benchmarks: AlpacaEval 2.0 (Dubois et al., 2024), Arena-Hard (Li et al., 2024), and MT-Bench (Zheng et al., 2024), ADS exhibits generalized performance enhancement, even enabling the relative smaller 7B/9B LLMs to compete with the larger 72B/27B counterparts. It is worth noting that across tasks of varying categories and complexities, ADS typically improves performance in knowledge, reasoning, and challenging tasks. To sum up, we highlight our contributions as follows:

- We pioneer the idea of automating the data search process for training LLMs, a task currently handled by expert human efforts, making a further step towards fully automated self-improving AI systems capable of continuous learning and adaptation.

- We propose the ACTIVE DATA SEARCH (ADS) framework, which encapsulates existing data collection and synthesis methods into APIs, enabling LLMs to generate optimal API calls for efficient data discovery through interactions with diverse environments.

- We showcase the effectiveness of ADS by conducting comprehensive experiments on 1,000 diverse in-house test tasks and three public benchmarks. Our results demonstrate that LLMs equipped with ADS exhibit substantial performance improvements. Furthermore, we provide an in-depth analysis of the underlying factors contributing to the efficacy of ADS.

## 2 RELATED WORK

**LLM as an Optimizer** With the progressive advancement of LLMs, LLM-as-an-optimizer (i.e., maximizing a downstream metric of an AI system using LLMs without human intervention) has emerged as a new paradigm, where the input optimization task and the output solution are described in natural language and processed by an LLM. This approach has been pioneered in automatic prompt engineering (Pryzant et al., 2023; Zhou et al., 2023; Wang et al., 2024; Guo et al., 2024; Yang et al., 2024b; Xiao et al., 2024), pipeline optimization of LLM-based agents (Zhang et al., 2024; Khattab et al., 2024; Yuksekgonul et al., 2024; Zhuge et al., 2024; Zhou et al., 2024b), and new algorithm discovery (Liu et al., 2024a; Lu et al., 2024). Our research distinguishes itself from these previous studies as our optimization goal is to enhance the fundamental capabilities of LLMs through automated data collection and curation, which is very important given that data is the (perhaps most) major power engine for today's LLMs. Moreover, existing work simply prompts off-the-shelf LLMs for optimization, which can lead to sub-optimal performance as these LLMs have not been explicitly trained to do so. Conversely, we develop methods to fine-tune the optimizer models.

**Synthetic Data Generation** Synthetic data, which leverages the generation capability of LLMs for data construction, has become crucial components across various stages in LLMs development (Long et al., 2024), including pre-training (Gunasekar et al., 2023; Li et al., 2023; Ben Allal et al., 2024), supervised fine-tuning (Taori et al., 2023; Mukherjee et al., 2023; Xu et al., 2023), and preference learning (Rafailov et al., 2024; Dubey et al., 2024; Adler et al., 2024). Instead of contributing to the booming spectrum by proposing yet another data synthesis method, we introduce a meta-optimization framework, which aims to integrate different data synthesis techniques and develop the best strategy (our optimizer) to leverage them. More recently, there is also a growing interest in targeted data synthesis for enhancing the model's particular weaknesses (Lee et al., 2024; BAAI, 2024; Cheng et al., 2024). Nevertheless, these approaches require extensive evaluation to spot the weaknesses, which sometimes necessitates expert-level knowledge for test set design and

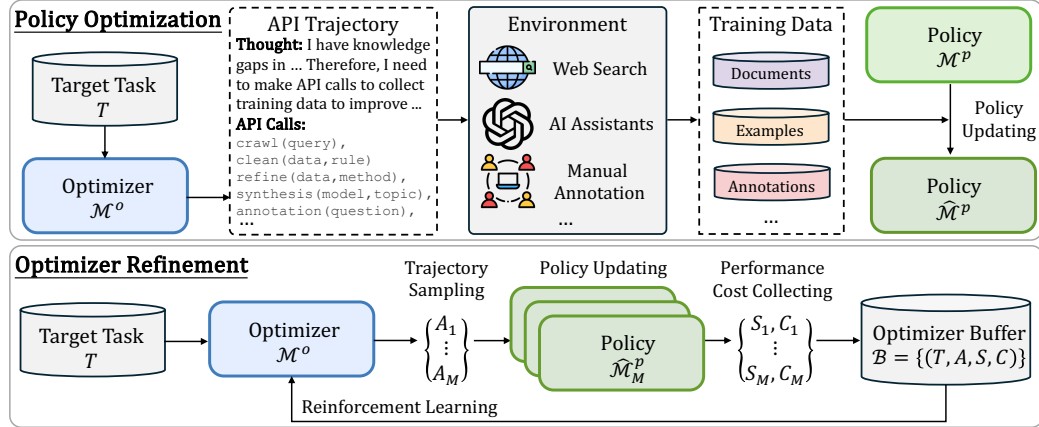

Figure 1: Overview of ACTIVE DATA SEARCH (ADS) framework. Upper: The optimizer is developed to generate optimal API calls and execute them in the environment for training data acquisition. The policy is then updated using the collected training data. Bottom: To make effective and efficient data-collecting decisions, the optimizer is iteratively refined through reinforcement learning.

result analysis. In contrast, our work strives to develop self-knowledge capabilities; the AI system knows its strengths and areas for improvement without any external feedback or guidance. Specifically, given only a small set of representative task queries, the optimizer introspects and identifies potential capability gaps that prevent the policy from completing the target task, and strategically generates optimal API calls to collect training data for addressing the identified gaps.

**Self-Improving of LLMs** Developing an autonomous self-improving system has been a major focus of current LLM research. Conventional reinforcement learning from human feedback (RLHF) methods (Ziegler et al., 2019; Stiennon et al., 2020; Ouyang et al., 2022; Dong et al., 2024; Liu et al., 2024b; Rosset et al., 2024) train a reward model from human preference data and subsequently use this model to fine-tune LLMs via reinforcement learning. The reliance on human preference data of course can be removed for scenarios where the correctness of output can be validated automatically and unambiguously (Zelikman et al., 2022; Gulcehre et al., 2023; Singh et al., 2023). For general instruction following, recent studies have built upon the success of LLM-as-a-judge (Zheng et al., 2024) and leverage AI feedback for self-improvement, an approach also known as RLAIF (Bai et al., 2022b; Lee et al., 2023). Yuan et al. (2024) and Wu et al. (2024) have further advanced this concept by employing LLMs to provide self-rewarding feedback on their own generated outputs. Unlike previous work that constrains the system from leveraging external signals, our framework allows for information inflow from external environments through the interactions of API calls, potentially elevating the upper bounds of self-improving.

## 3 SELF-IMPROVING VIA ACTIVE DATA SEARCH

The primary objective of our ACTIVE DATA SEARCH (ADS) framework is to endow AI systems with the capability to acquire valuable training data from external environments for retraining themselves, thereby facilitating self-improvement with minimal human supervision. As illustrated in Figure 1, we develop an *optimizer* model for a *policy* model. The optimizer model is designed to discover tailored training datasets to improve the policy by generating textual API calls to interact with various resources and environments. In this section, we start with a brief introduction to the problem formulation, followed by an overall description of policy optimization and optimizer refinement processes. We defer the details of our prototype implementations in Section 4.

### 3.1 PROBLEM FORMULATION

Given a target task $T$ and a policy model $\mathcal{M}^p$, the goal of ADS is to maximize the performance of the policy model on $T$ through active search for optimal training data. The data search process is conducted by an optimizer model $\mathcal{M}^o$ and a set of pre-defined APIs. Each API requires some input parameters and returns output data gathered from various environments. These APIs can be implemented through a variety of established methods, which may encompass but are not limited to data crawling, filtering, cleaning, refinement, synthesis, and manual annotation. The resultant data can

also be sourced from a broad spectrum, such as internet search engines, LLM-based AI assistants, or crowdsourcing annotation services. To mitigate excessive and unnecessary API invocations, we assign varying costs to different APIs according to their practical complexities. Following common practice, each target task is represented by a set of instructions $Q$, which serve as input for the optimizer model. The output of this model is an action trajectory $A$, comprising multiple API calls.

## 3.2 POLICY OPTIMIZATION

The system optimizes its policy model $\mathcal{M}^p$ for a given target task $T$ through the following steps. First, the optimizer model $\mathcal{M}^o$ is instructed to conduct a comprehensive analysis of the instructions $Q$, identifying the essential required capabilities. Following this, $\mathcal{M}^o$ performs an introspective assessment of its own proficiencies and deficiencies on these identified capabilities, and generates sequential API calls $A = [a_1, a_2, ..., a_K]$ for data acquisition, where $K$ is the number of API calls and is determined on-the-fly. Notably, the above process is completed by the optimizer as free-form text generation, enabling seamless incorporation of any pre-defined APIs.

Next, we execute each API call $a_k$ in the environment $\mathcal{E}$, and collect the returned data to obtain a training dataset $D$, which may include a variety of data types such as raw documents, supervised demonstrations, and preference data, following the best practice of current LLM training. Finally, we train the original $\mathcal{M}^p$ on the tailored training dataset to update its knowledge and capacities, resulting in an updated policy model $\hat{\mathcal{M}}^p$.

## 3.3 OPTIMIZER REFINEMENT

To make efficient data-collecting decisions, the optimizer model $\mathcal{M}^o$ should be aware of the strengths and weaknesses of the policy model $\mathcal{M}^p$. Therefore, we propose an iterative training process for optimizer refinement. First of all, we assume a set of training tasks $\mathcal{T} = \{T\}$, where reliable performance measurements are available. It is important to note that we *do not* make this assumption at test-time since the evaluation of arbitrary real-world tasks can be complex and resource-intensive. Instead, we posit that the self-knowledge developed for the optimizer during the training phase can be effectively generalized to novel test tasks.

We then employ reinforcement learning and consider both performance gains and API costs in reward design. The training is conducted iteratively. At each iteration, we sample multiple API trajectories $\mathcal{A} = \{A\}$ for each training task $T$. We then execute the data acquisition and policy training processes accordingly and collect the corresponding performance gain $S(A)$ and API cost $C(A)$. To avoid potential data leakage, we partition the instruction set $Q$ of $T$ into an observed set $Q^o$ and another held-out set $Q^h$. The optimizer can only see the examples in the observed set and the performance gains are measured based on the held-out set. This results in a set of quadruple $\mathcal{B} = \{(T, A, S, C)\}$. We then optimize $\mathcal{M}_t^o$ to $\mathcal{M}_{t+1}^o$ on $\mathcal{B}$ with reinforcement learning algorithms such as rejection sampling (Bai et al., 2022a) and direct preference optimization (Rafailov et al., 2024). The complete process of optimizer refinement is described in Algorithm 1.

---

**Algorithm 1** Optimizer Refinement Procedure in ADS

---

**Require:** Training tasks $\mathcal{T}$, policy model $\mathcal{M}^p$, optimizer model $\mathcal{M}^o$, environment $\mathcal{E}$.
1: **for** iteration $t$ in $N$ **do**
2:     Initialize optimizer buffer $\mathcal{B} = \{\}$.
3:     **for** task $T$ in $\mathcal{T}$ **do**
4:         Split task instructions $Q$ into observed set $Q^o$ and held-out set $Q^h$.
5:         Sample API trajectories $\mathcal{A} \sim \mathcal{M}_t^o(\cdot|Q^o)$.
6:         **for** trajectory $A$ in $\mathcal{A}$ **do**
7:             Execute $A$ in $\mathcal{E}$ to acquire data to update $\mathcal{M}^p$ to $\hat{\mathcal{M}}^p$.
8:             Collect performance gain $S(A)$ of $\hat{\mathcal{M}}^p$ on $Q^h$, and API cost $C(A)$.
9:             Add $(T, A, S, C)$ to $\mathcal{B}$.
10:        **end for**
11:     **end for**
12:     Optimize $\mathcal{M}_t^o$ to $\mathcal{M}_{t+1}^o$ using reinforcement learning on $\mathcal{B}$.
13: **end for**
14: **return** $\mathcal{M}_N^o$.

---

| API Name | Parameter | Return | Description | Cost |
|---|---|---|---|---|
| `Information Retrieval` | Query | Document | Sparse and Dense Retrieval | Low |
| `Demonstration Generation` | Topic | Instruction-Response Pair | Synthesis with the Policy | Medium |
| `Question Answering` | Question | Question-Answer Pair | Annotated by Stronger LLM | High |

Table 1: Implementation details of our data collecting APIs.

## 4 EXPERIMENTAL SETUP

### 4.1 APIs

We design three distinct data-collecting APIs to facilitate the *acquisition*, *utilization*, and *enhancement* of knowledge. The first API is `Information Retrieval`, which employs both sparse and dense retrieval to retrieve relevant documents from external knowledge databases such as Wikipedia, thereby supporting knowledge acquisition, analogous to the pre-training stage of LLMs. The second API is `Demonstration Generation`, which utilizes the policy model to generate appropriate exemplar instruction-response pairs, tailored to various knowledge application scenarios, reminiscent of the alignment stages of LLMs. The last API is `Question Answering`, which resorts to the wisdom of human experts, mimicking how humans learn from each other. [1]. It provides to-the-point answers to questions proposed by the optimizer model. Notably, this framework is not restrictive and can accommodate additional APIs as needed. In each API trajectory, these APIs are invoked sequentially, e.g., `api_name_1(api_param_1)...api_name_n(api_param_n)`. Subsequently, the corresponding API calls are executed to formalize the training dataset. We then combine the training datasets from different API calls to update the policy model. For tasks where the model possesses the required capabilities, a "none" option is incorporated. We present the API implementation details in Table 1.

To avoid excessive API calls, e.g., asking questions that the policy model already knows the answer to, we take into account the API cost in our implementation. The cost associated with each API varies according to their practical complexities and computational overhead. Concretely, we assign relative costs as follows: `Information Retrieval` is at one since retrieval is fast and cheap, `Demonstration Generation` at two given generation is slow and resource-intensive, and `Question Answering` at three as it requires a more powerful model or manual efforts. Further implementation details of APIs are provided in Appendix A.

### 4.2 DATASET

To train and evaluate the optimizer model, it is essential to collect a diverse set of "target tasks" for which we aim to optimize the performance of the policy model. For this purpose, we adopt the general instruction-following dataset Llama-3-Magpie-Air-3M-v0.1 (Xu et al., 2024). Each instruction in this dataset is associated with several labels such as task category, intent, and difficulty. The instructions and labels are generated by Llama-3-8B-Instruct (Dubey et al., 2024). Subsequently, we group the instructions with identical labels into different clusters, with each cluster representing a distinct "target task". We discard tasks with fewer than five instructions. The resulting dataset comprises 10,239 distinct tasks, which we partition into 8,739 tasks for training, 500 tasks for validating, and 1,000 tasks for testing. In the train and valid splits, we allocate three observed instructions per task for trajectory generation and reserve two held-out instructions for performance measurement. This approach ensures that the optimizer learns to improve performance at *task level* rather than *instance level*, as over-fitting on the observed instructions may not necessarily translate performance gains on the held-out instructions. For the test split, to enhance the robustness of evaluation, we augment the initial five instructions per task to 100 using Self-Instruct (Wang et al., 2023), with three instructions as observed and 97 as held-out. More details of our dataset are shown in Appendix B.

### 4.3 EVALUATION METHODS

**In-house Evaluation** We employ both the RM and GPT-4 (Achiam et al., 2023) to compare responses generated by the original and updated policy models on held-out instructions across our test tasks. We report the average win, tie, and lose rate across all tasks. For RM evaluation, we sample five API trajectories and compute the average RM score among them, ensuring statistical robustness. For GPT-4 judgment, we employ a head-to-head comparison using the pairwise evaluation prompt

---

[1]For efficiency and reproducibility, we employ a stronger LLM as a proxy for human supervision in practice.

proposed by Zheng et al. (2024). We try two different response orders to prevent order sensitivity. To save token usage, we only sample one trajectory and test on two held-out instructions per task.

**Generalizing to Public Benchmarks** We additionally utilize three well-established benchmarks for further evaluating the generalizability of the trained optimizer models, including AlpacaEval 2.0 (Dubois et al., 2024), Arena-Hard (Li et al., 2024), and MT-Bench (Zheng et al., 2024). AlpacaEval 2.0 includes 805 instructions for daily chat scenarios while Arena-Hard contains 500 more challenging real-world questions. MT-Bench consists of 80 multi-turn dialogues spanning eight distinct domains[2]. Since the original version of MT-Bench contains incorrect reference responses, we follow previous works (Adler et al., 2024; Wan et al., 2024) to use the updated version for evaluation. The proposed approach aims to optimize the performance on specific target tasks. However, these benchmarks (AlpacaEval 2.0, Arena-Hard, and MT-Bench) cover a wide range of distinct tasks, which can be too broad to be considered as a single target task. To conduct meaningful experiments, we make the following adaptations. First, we cluster the original instructions based on task category and difficulty, utilizing the approach in Xu et al. (2024). This process yields 38, 34, and 21 "target tasks" for AlpacaEval 2.0, Arena-Hard, and MT-Bench, respectively. Then, we employ the Self-Instruct approach to generate three new instructions for each task and use them as the observed instructions for that task. We only assess model performance on the original instructions provided in these benchmarks. Following conventions (Meng et al., 2024; Adler et al., 2024), we conduct a standard evaluation using GPT-4 judgment. We adhere to the default setups and report the win rate of the updated policy models against the GPT-4 baseline for each benchmark. Additionally, we conduct a supplementary evaluation using RM as judgment and report the average adjusted win rate, calculated as win rate $+ 0.5 \times$ tie rate. For both RM and GPT-4 judgments, we also report the weighted averages of these benchmarks based on their number of samples. Further details of the evaluation process can be found in Appendix D.

### 4.4 TRAINING DETAILS

We employ Qwen-2-7B-Instruct (Yang et al., 2024a) and Gemma-2-9B-Instruct (Riviere et al., 2024) as our policy models[3]. The optimizer model is initialized using the corresponding policy model.

During each iteration of ADS, the optimizer model generates five API trajectories per target task, based on the observed instructions. To further enhance the diversity of the trajectories, we incorporate three additional default trajectories, each corresponding to a specific API type. These trajectories are executed in the data-collecting environments to construct a comprehensive training dataset. Next, the policy model is updated through either in-context learning or fine-tuning using this training data. Considering the frequent policy model updating for each target task, we use in-context learning to maintain computational efficiency while preserving performance, since extensive research has demonstrated that in-context learning can achieve comparable effectiveness to fine-tuning (Mosbach et al., 2023; Agarwal et al.). We then determine the reward signal of each API call trajectory by evaluating the performance gains of the updated policy model on the held-out instructions. We use FsfairX-Llama-3-RM-v0.1 (Xiong et al., 2024) as the RM for evaluation. Finally, we iteratively update the optimizer model by direct preference optimization (DPO) (Rafailov et al., 2024) on the chosen and rejected trajectories with the highest and lowest rewards. To ensure training stability, we perform warm-up rejection sampling (RS) (Bai et al., 2022a) on the chosen trajectories exhibiting the highest rewards before iterative DPO for optimizer model training. In Section 6.1, we further illustrate the comparison between DPO and RS, as well as investigate the impact of the default API trajectories. The iterative training process of the optimizer model is presented as follows:

- Prompting: $\mathcal{M}^o$ is initialized from $\mathcal{M}^p$ without fine-tuning, then direct prompting.
- RS Iteration 0: $\mathcal{M}^o_0$ is initialized from $\mathcal{M}^o$, then warm-up RS on chosen API trajectories.
- DPO Iteration 1, 2, 3: $\mathcal{M}^o_1$, $\mathcal{M}^o_2$, $\mathcal{M}^o_3$ are initialized from $\mathcal{M}^o_0$, $\mathcal{M}^o_1$, $\mathcal{M}^o_2$, then DPO on chosen and rejected API trajectories.

To enhance the optimizer model's capacity for reward improvements while minimizing API costs, we introduce a novel cost-control approach, which draws inspiration from the length-control strategy

---

[2]Given that our dataset focuses on single-turn tasks, we evaluate the 1st turn performance for consistency.

[3]Since our training data are generated by Llama-3-8B-Instruct, we exclude it from our choices of policy models to avoid any potential biases.

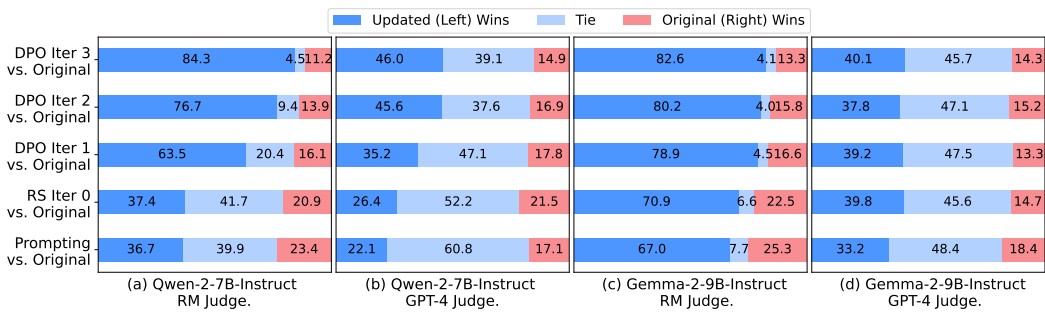

Figure 2: Comparisons between responses generated by the updated and original policy models across in-house test tasks with both RM and GPT-4 judgments. The updated policy models are developed on training data collected by different iterations of optimizer models.

employed in LLM alignment (Wu et al., 2024). Our approach posits that among trajectories with comparable rewards, those associated with lower costs are more valuable for optimization and thus deserve higher reinforcement feedback. Specifically, we introduce a cost tier parameter $\tau \in [0, 1]$ to control the trade-off between rewards and costs. Trajectories within the top-tier rewards ranging $[(1 - \tau)R_{\max} + \tau R_{\min}, R_{\max}]$ are considered to have similar performance. From this subset, we select the trajectory with the lowest cost as the chosen trajectory. Conversely, for the reject trajectory, we select the one with the highest cost within $[R_{\min}, (1 - \tau)R_{\min} + \tau R_{\max}]$. To maintain an optimal balance between reward and cost, we empirically determine the cost tier parameter at 0.1. We further conduct experiments to illustrate the effectiveness of the proposed cost-control approach in Section 6.2. More information on training details can be found in Appendix C.

## 5 MAIN RESULTS

### 5.1 IN-HOUSE EVALUATION

Figure 2 illustrates the evaluation results on target tasks across in-house test split. We compare the responses generated by the updated and original policy models, displaying the task win, tie, and lose rates with both RM and GPT-4 judgments.

**ADS presents significant superiority across 1,000 in-house test tasks.** We can first observe that the prompting method yields slight performance gains. For instance, the updated policy model achieves a win rate of 36.7% compared to the original policy model's 23.4% in RM judgment, and 22.1% to 17.1% in GPT-4 judgment for Qwen-2-7B-Instruct. Following the iterative training process, the final model exhibits a remarkable enhancement, with a win rate of 84.3% versus 11.2% in RM evaluation and 46.0% versus 14.9% in GPT-4 evaluation. These substantial performance improvements can be attributed to the valuable training data discovered by the optimizer model in ADS, which enhances the knowledge and capabilities required to accomplish these test tasks.

**Iterative ADS boosts consistent performance improvements.** During the initial iteration of DPO training, our optimizer model exhibits rapid adaptation to high-reward API trajectories. Consequently, the updated policy model achieves substantial improvements in win rates: from 37.4% to 63.5% according to RM judgment, and from 26.4% to 35.2% as evaluated by GPT-4 for Qwen-2-7B-Instruct. As the number of training iterations increases, the updated policy model consistently demonstrates improved win rates. Given that the training tasks remain constant across different iterations, we posit that the consistent performance gains are from the automatic weakness identification and refinement during the iterative training process, thus progressively increasing the probabilities of generating optimal API trajectories for self-improvement.

### 5.2 PUBLIC BENCHMARKS

In Table 2, we present the evaluation results on three public benchmarks, including AlpacaEval 2.0, Arena-Hard, and MT-Bench. We compare the responses generated by the policy model and the GPT-4 baseline model and show the win rates with both RM and GPT-4 judgments.

| Models | RM Judgment | | | | GPT-4 Judgment | | | |
|---|---|---|---|---|---|---|---|---|
| | AE | AH | MT | Avg. | AE | AH | MT | Avg. |
| Qwen-2-7B-Instruct ($\mathcal{M}^p$) | 31.7 | 56.1 | 66.9 | 42.6 | 24.0 | 25.6 | 55.9 | 26.4 |
| *Prompting ($\mathcal{M}^o \rightarrow \mathcal{M}^p$)* | 33.2 | 57.0 | 68.1 | 43.8 | 24.9 | 26.7 | 52.5 | 27.2 |
| *RS Iteration 0 ($\mathcal{M}_0^o \rightarrow \mathcal{M}^p$)* | 32.4 | 56.7 | 68.1 | 43.2 | 24.2 | 26.7 | 53.8 | 26.8 |
| *DPO Iteration 1 ($\mathcal{M}_1^o \rightarrow \mathcal{M}^p$)* | 34.9 | 57.9 | 66.9 | 45.0 | 28.6 | 28.0 | 53.8 | 29.8 |
| *DPO Iteration 2 ($\mathcal{M}_2^o \rightarrow \mathcal{M}^p$)* | 36.5 | 60.6 | 69.4 | 47.1 | 30.6 | 28.8 | 57.7 | 31.5 |
| *DPO Iteration 3 ($\mathcal{M}_3^o \rightarrow \mathcal{M}^p$)* | **38.8** | **61.1** | **71.9** | **48.8** | **31.9** | **30.1** | **59.4** | **32.8** |
| $\Delta$ *to Qwen-2-7B-Instruct* | (+22.3%) | (+8.9%) | (+7.5%) | (+14.6%) | (+32.9%) | (+17.7%) | (+6.2%) | (+24.3%) |
| Gemma-2-9B-Instruct ($\mathcal{M}^p$) | 28.6 | **70.0** | **71.3** | 46.0 | 34.8 | **37.5** | 55.0 | 36.9 |
| *Prompting ($\mathcal{M}^o \rightarrow \mathcal{M}^p$)* | 28.0 | 59.2 | 70.0 | 41.7 | 33.8 | 30.0 | 56.3 | 33.7 |
| *RS Iteration 0 ($\mathcal{M}_0^o \rightarrow \mathcal{M}^p$)* | 30.5 | 65.6 | **71.3** | 45.5 | 35.3 | 34.8 | 51.3 | 36.0 |
| *DPO Iteration 1 ($\mathcal{M}_1^o \rightarrow \mathcal{M}^p$)* | 30.4 | 48.4 | 66.9 | 39.0 | 36.7 | 22.4 | 54.4 | 32.6 |
| *DPO Iteration 2 ($\mathcal{M}_2^o \rightarrow \mathcal{M}^p$)* | **33.0** | 67.9 | 70.6 | **47.8** | 36.2 | 35.2 | **57.5** | 37.1 |
| *DPO Iteration 3 ($\mathcal{M}_3^o \rightarrow \mathcal{M}^p$)* | 32.8 | 65.7 | 69.4 | 46.8 | **37.0** | 37.4 | 56.9 | **38.3** |
| $\Delta$ *to Gemma-2-9B-Instruct* | (+14.5%) | (-6.1%) | (-2.6%) | (+1.6%) | (+6.3%) | (-0.2%) | (+3.4%) | (+3.7%) |
| Llama-3.1-70B-Instruct | 53.0 | 76.7 | 83.8 | 63.3 | 39.5 | 57.0 | 75.0 | 47.9 |
| Qwen-2-72B-Instruct | 38.9 | 66.8 | 73.1 | 50.9 | 35.1 | 48.3 | 66.9 | 41.7 |
| Gemma-2-27B-Instruct | 33.6 | 80.2 | 75.0 | 52.8 | 37.0 | 49.9 | 60.6 | 43.0 |

Table 2: Comparisons between responses generated by the updated policy models and the baseline GPT-4 on AlpacaEval 2.0 (AE), Arena-Hard (AH), and MT-Bench (MT) with both reward model (RM) and GPT-4 judgment. Percentages indicate the relative improvements/decreases observed in the final iteration of the updated policy model when compared to the original policy model.

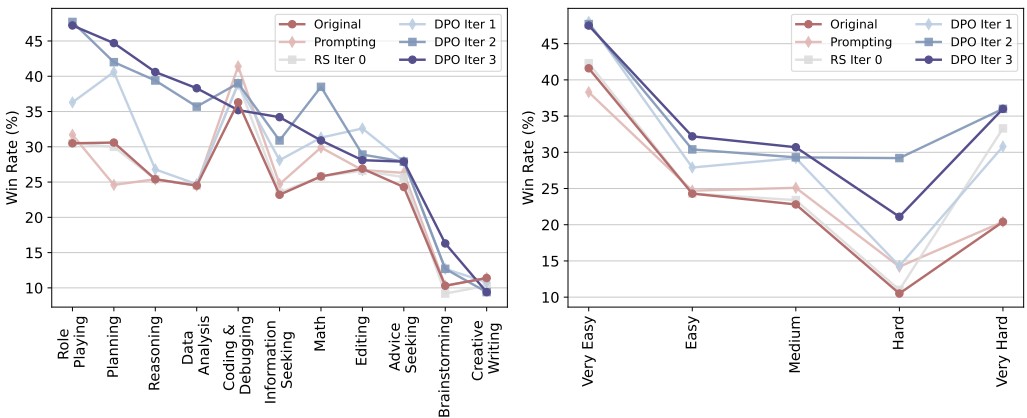

Figure 3: Fine-grained evaluation of different updated policy models and the original Qwen-2-7B-Instruct across various task categories (left) and difficulties (right) in AlpacaEval 2.0.

**ADS demonstrates generalized capabilities on public benchmarks.** For the target tasks from public benchmarks, our approach exhibits generalized improvements. Overall, the updated policy model of Qwen-2-7B-Instruct shows significant relative gains across all benchmarks, with an average enhancement of +14.6% and +24.3% in RM and GPT-4 judgment, respectively. Notably, in AlpacaEval 2.0, the improvements are even more pronounced, e.g., +22.3% in RM evaluation and +32.9% in GPT-4 evaluation. These findings reveal that our optimizer model, despite being trained on limited target tasks, demonstrates the capacity to find valuable training data for practical tasks, effectively addressing the challenges for real-world task optimization.

**ADS enhances policy model to rival that of more powerful LLMs.** We show that with the help of training data collected from our optimizer model, smaller and weaker policy models can achieve comparable results to those of larger and stronger LLMs. Specifically, in the RM evaluation within AlpacaEval 2.0, Qwen-2-7B-Instruct with ADS achieves a win rate of 38.8%, which is almost equivalent to the performance of the substantially larger Qwen-2-72B-Instruct. Similarly, in the GPT-4 evaluation, Gemma-2-9B-Instruct with ADS, achieves a win rate of 37.0%, matching the performance of larger Gemma-2-27B-Instruct.

**ADS typically enhances knowledge-intensive and reasoning-related tasks.** We conduct a fine-grained evaluation across distinct task categories in AlpacaEval 2.0. The results of Qwen-2-7B-

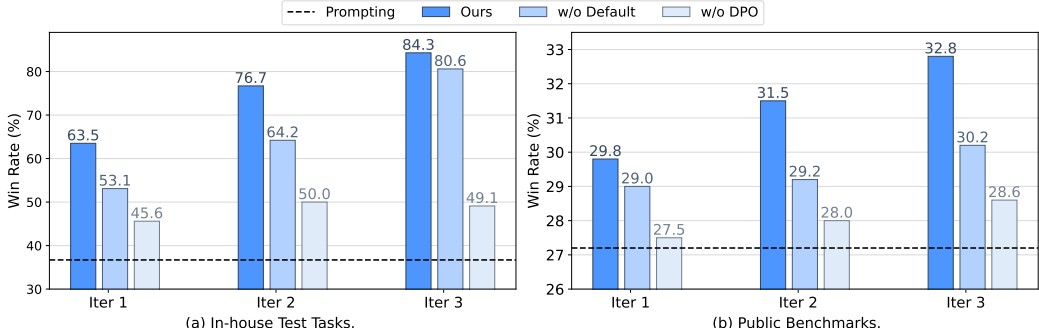

Figure 4: Ablation study results of Qwen-2-7B-Instruct across in-house test tasks with RM judgment (left) and public benchmarks with GPT-4 judgment (right), where "w/o DPO" refers to the replacement of direct preference optimization with rejection sampling algorithm, while "w/o Default" denotes the exclusion of default API trajectories for optimizer model training.

Instruct are illustrated in Figure 3 (left). We find that compared to the original baseline, ADS improves the performance in most of the categories, especially for those that require substantial general or specific knowledge and advanced reasoning abilities, such as information-seeking, role-playing, reasoning, and planning. However, in categories like editing, creative writing, and coding & debugging, our approach only has slight improvements or maintains comparable to the baseline. This limited enhancement can be potentially attributed to the inherent nature of these tasks, which primarily involve format and style rewriting, as well as fragment modifications, presenting inherent challenges for optimization through in-context learning from acquired training data.

**ADS particularly improves in complex and challenging tasks.** In Figure 3 (right), we further demonstrate the performance across tasks with various levels of difficulty. The results reveal that ADS yields more substantial improvements as task complexity increases. Specifically, the relative performance enhancement increases from +14.2% for very easy tasks to a remarkable +76.5% for tasks categorized as very hard. These results further demonstrate the potential of ADS for continuous self-improving, typically in complex tasks that lack expert supervision.

# 6 ABLATION AND ANALYSIS

## 6.1 ABLATION STUDIES

In this section, we delve into the key components in our practical implementation of ADS, focusing on the training algorithm and the construction of API trajectories. We first investigate and compare the performance of the optimizer model when trained iteratively using rejection sampling and direct preference optimization. Subsequently, we analyze the influence of adding default API trajectories into the optimizer model's training process. The evaluation results across in-house test tasks with RM judgment and public benchmarks with GPT-4 judgment are shown in Figure 4.

**DPO enhances discrimination between chosen and rejected trajectories.** We can observe that compared to rejection sampling, the implementation of direct preference optimization substantially improves the optimizer model's capacity to differentiate between chosen and rejected API trajectories, which is illustrated by a significant increase in the average win rate from 49.1% to 84.3% across in-house test tasks. Similarly, in public benchmarks, the win rate improved from 28.6% to 32.8%. This enhanced discriminative capability facilitates more effective weakness identification and decision-making processes of our optimizer model, ultimately leading to high-reward API trajectory exploration and optimal training data acquiring for deficiency enhancement.

**Incorporating default API trajectories enhances trajectory diversity.** In addition to the five trajectories sampled by the optimizer during its training process, we incorporate three default trajectories, each corresponding to a distinct data-collecting API in our implementation. Table 4 in Appendix C illustrates an example of both default and generated trajectories. The inclusion of these default API trajectories alongside the self-generated candidates results in a more diverse set of trajectories for optimizer model training, facilitating improved average win rates across all iterations, both in in-house test tasks and public benchmarks.

## 6.2 ANALYSIS RESULTS

We further make an in-depth analysis to investigate the factors contributing to the efficacy of ADS, focusing on two key aspects: the advantages of self-explored training data and the implementation of the cost-control mechanism. To evaluate the impact of self-explored training data, we compare the API trajectories generated by our optimizer model against those curated using a baseline strategy. This baseline approach utilizes the `Question Answering` API for each observed instruction in the target task to construct the corresponding API trajectory. Subsequently, we employ the collected data for policy model optimization in both cases. Regarding the cost-control mechanism, we compare our method with an alternative approach that prioritizes reward maximiza-

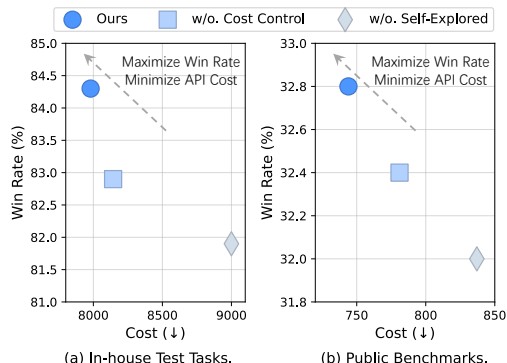

Figure 5: Effectiveness of self-explored training data and cost-control mechanism in maximizing win rate while minimizing API cost.

tion without considering API costs. Figure 5 illustrates these analytical results, encompassing both in-house test tasks using RM judgment (left) and public benchmarks with GPT-4 judgment (right).

**Self-explored data presents more suitable for LLMs training.** Despite employing the most sophisticated and expensive `Question Answering` API for training data construction, the approach without self-explored training data achieves a relatively lower win rate, e.g., 81.9% versus 84.3% across in-house test tasks, while incurring higher API costs, e.g., 9,000 versus 8,219, compared to our approach. The optimizer model in ADS is designed to automatically identify and address the potential capability gaps in specified tasks based on the developed self-knowledge. Consequently, our approach provides a more tailored and targeted performance improvement.

**Cost-control mechanism reduces the API cost while improving performance.** To maximize the potential for self-improvement while maintaining appropriate resource allocation, we implement a cost-control mechanism that optimizes the trade-off between minimizing costs and maximizing performance during the training process of the optimizer model, as detailed in Section 4.4. In comparison to the approach that focuses solely on maximizing performance without considering costs, our method not only achieves a lower cost as expected but also demonstrates an improved win rate. The performance improvements can be attributed to the increased trajectory diversity compared to the reward maximization approach. This observation indicates that our approach enables the development of a robust and cost-effective system for data acquisition, ultimately contributing to improved overall performance and reduced computational overhead.

## 7 CONCLUSION & LIMITATIONS

In this study, we explored enabling LLMs to autonomously acquire optimal training data for self-improvement with minimal human intervention. We proposed a novel framework, ACTIVE DATA SEARCH (ADS), which utilizes LLMs themselves as an optimizer to strategically invoke appropriate APIs, facilitating the discovery of tailored training datasets from external environments for self-training. To ensure efficient data-collecting decisions, we introduced an iterative refinement algorithm for the optimizer, guided by reinforcement feedback signals aiming to maximize task performance while minimizing computation costs. Through a series of experiments on 1,000 in-house test tasks and three public benchmarks, we demonstrated the effectiveness and generalizability of ADS. Notably, ADS exhibited the capacity to enhance the performance of smaller and weaker language models to be comparable with larger and stronger LLMs on AlpacaEval 2.0. This automated process of data discovery and self-training presents opportunities to reduce the reliance on human expertise in LLM development, providing a new direction for future research in this domain.

The current implantation is a proof-of-concept with several limitations. First, the optimizer and policy models are separate. Unifying them is an interesting avenue for future work. Second, the APIs included currently are far from covering all existing data techniques. Third, we did not consider multi-turn optimization of the policy models in the framework.

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

# A   DETAILS OF APIS

In this section, we first present the prompt for API trajectory generation of the optimizer model in Figure 6, then show the detailed implementation of our data collecting APIs, including (1) `Information Retrieval`, which facilitates efficient knowledge acquisition; (2) `Demonstration Generation`, which enables various knowledge utilization scenarios; and (3) `Question Answering`, which serves to enhance and refine the acquired knowledge.

---

**Prompt for API Trajectory Generation**

# Task Overview
Your goal is to analyze input prompts, identify knowledge gaps, and strategically use provided APIs to enhance your knowledge and capabilities.

# Provided APIs
You will have access to the following APIs to obtain additional data for improvements:
1. "information_retrieval(query: string)": Retrieves relevant documents for a given search query. Use for factual knowledge gaps.
2. "example_instantiation(topic: string)": Generates practical instances based on a given topic. Use for applying knowledge to concrete situations.
3. "question_answering(question: string)": Provides answers to a given question. Use for deeper understanding of knowledge.

# Constraints and Guidelines
1. Focus on common and general knowledge and capabilities requirements across all the input prompts.
2. Use API calls only to address competence gaps when necessary, not to directly solve the prompts.
3. If you need to make API calls, formatted as <api_calls><api>api_name_1(api_param_1) </api><api>api_name_2(api_param_2) </api>...</api_calls>
4. If you do not need to make API calls, formated as <api_calls>none</api_calls>

# Input:
{observed_instructions}

---

Figure 6: Prompt for API trajectory generation of the optimizer model. {observed_instructions} is the placeholder for the observed instructions in the target task.

**Information Retrieval**   For our retrieval corpus, we utilize the Wikipedia (en) from December 20th, 2022 [4], encompassing approximately 8.59 million paragraphs. Our retrieval process employs a two-stage approach. Initially, we implement sparse retrieval using the BM25 [5] algorithm to identify the top 1,000 most relevant documents for a given search query. Subsequently, we refine this selection through dense retrieval, leveraging the BGE-Large-EN-v1.5 (Xiao et al., 2023) embedding model to obtain the most appropriate document from the previously identified candidates.

**Demonstration Generation**   To obtain knowledge utilization examples, we employ an approach that relies solely on the policy model itself without incorporating external tools. Specifically, we leverage the policy model to synthesize demonstrations by generating instructions and corresponding responses based on a given knowledge topic. The prompt used for demonstration generation is shown in Figure 7.

**Question Answering**   To enhance the comprehension of acquired knowledge, we employ Llama-3.1-70B-Instruct (Dubey et al., 2024) as a replacement for human experts to generate comprehensive responses to complex questions. The prompt used for question answering is illustrated in Figure 8.

---

[4] https://huggingface.co/datasets/Cohere/wikipedia-22-12
[5] https://github.com/xhluca/bm25s

---

**Prompt for Demonstration Generation**

Generate an Instruction and the corresponding comprehensive Response related to the Topic.

Topic: {topic}

---

Figure 7: Prompt for Demonstration Generation. {topic} is the placeholder for the knowledge topic.

---

**Prompt for Question Answering**

Provide a detailed Answer to the given Question.

Question: {question}

---

Figure 8: Prompt for question answering. {question} is the placeholder for the given question.

## B  DETAILS OF DATASET

We group instructions into clusters based on three key attributes: category, intent, and difficulty. The category attribute represents the broad task type, encompassing areas such as creative writing, reasoning, and coding. The intent attribute indicates the primary objectives within the instructions, including getting helpful tips, identifying logical fallacy, develop software extensions. The difficulty attribute quantifies the complexity of following the instructions, ranging from very easy to very hard.

To improve the robustness of evaluation results, we augment the size of the held-out set for each target task in the test split. Specifically, we leverage five existing instructions as seed examples, and prompt GPT-4 to generate new instructions that are significantly different from these initial examples but belong to the same task (Wang et al., 2023). To maintain distinctiveness, we employ a filtering mechanism whereby any generated instructions with a Rouge-L similarity score exceeding 0.7 when compared to the original instructions are eliminated. The prompt for instruction augmentation is illustrated in Figure 9. The statistics of our train, valid, and test splits are shown in Table 3.

| Statistics | Train | Valid | Test |
|---|---|---|---|
| # Task | 8,739 | 500 | 1,000 |
| # Category | 12 | 12 | 12 |
| # Intent | 4,982 | 371 | 902 |
| # Difficulty | 5 | 5 | 5 |
| # Obs. Inst. Per Task | 3 | 3 | 3 |
| # Held. Inst. Per Task | 2 | 2 | 97 |

Table 3: Statistics of train, valid, and test splits.

## C  DETAILS OF TRAINING PROCESS

For policy model updating, we employ ICL on the collected training dataset for each target task, including retrieved documents, instantiated instruction-response pairs, and answers to complex questions, as the supplementary information to address the given held-out instruction. The prompt for ICL of the policy model is shown in Figure 10.

For optimizer model updating, We train the optimizer model with a batch size of 128 and a maximum sequence length of 2048. The training is conducted on a single node with 8x80GB Nvidia A100 GPUs for one epoch per iteration. We perform RS for a single iteration and DPO for three iterations. The models are optimized using the AdamW (Loshchilov & Hutter, 2019) optimizer with $\beta_1 = 0.9$ and $\beta_2 = 0.999$. We use a weight decay of 0.0 and gradient clipping of 1.0. A cosine learning rate schedule is employed, with a warmup ratio of 0.1 and a maximum learning rate of 2e-5 for RS and 5e-7 for DPO. The $\beta$ parameter in DPO is set to 0.01. Our training framework is developed based on the HuggingFace Transformers (Wolf et al., 2020) and TRL (Werra et al., 2020). We show an example of default and generated API trajectories in Table 4.

---

**Prompt for Task-Specific Instructions Augmentation**

Given 5 instructions as demonstrations within a specific task, please generate 100 new distinct instructions that are relevant to the same task but significantly different from the demonstrations provided.

Task: {task}

Here are 5 demonstrations:
Instruction 1: {seed_instruction_1}
Instruction 2: {seed_instruction_2}
Instruction 3: {seed_instruction_3}
Instruction 4: {seed_instruction_4}
Instruction 5: {seed_instruction_5}

Now, please generate 100 new distinct instructions that are:

1. Significantly different from the demonstrations.
2. Within the same task or domain as the demonstrations.
3. Similar in complexity and difficulty to the demonstrations.
4. Varied in their approach to the task (e.g., different aspects or sub-tasks within the task).
5. Consistent with the tone and style of the demonstrations.

Please format your generated instructions as follows:

Generated Instruction 1: [Your generated instruction]
Generated Instruction 2: [Your generated instruction]
Generated Instruction 3: [Your generated instruction]
...

---

Figure 9: Prompt for task-specific instructions augmentation. {task} is the placeholder for the given task. {seed_instruction_i} is the placeholder for i-th seed instruction.

---

**Prompt for ICL**

{collected_data}

Your task is to read the above text to answer the following question.

1. Note that the above information is only visible to you, don't indicate that you have read above text in your answer. For example, don't say "according to the information provided above".
2. Do not solely rely on the provided information. You know much more than the provided information. Use it as a hint to write an comprehensive, informative, elaborate, and helpful answer.

{held_out_instruction}

---

Figure 10: Prompt for ICL of the policy model. {collected_data} is the placeholder for the training dataset derived from the optimizer model. {held_out_instruction} is the placeholder for the held-out instruction in the target task.

| **Example of Default and Generated API Trajectories** |
|---|
| **Observed Instructions:** 
 Explain the implications of the Einstein-Podolsky-Rosen paradox on quantum entanglement. 
 Describe the mathematical formulation of the wave function collapse in the Copenhagen interpretation. 
 Analyze the potential applications of quantum tunneling in developing next-generation transistors. |
| **Generated API Trajectory:** 
 # Analysis of Requirements 

 After analyzing the input prompts, I have identified the required background knowledge and broader capabilities as follows: 
 1. Advanced knowledge of quantum physics 
 2. Ability to explain complex scientific concepts 
 3. Understanding of quantum mechanics principles 
 4. Familiarity with practical applications of quantum theory 

 # Competence Assessment 

 I have knowledge gaps in quantum mechanics principles and capability gaps in explaining complex scientific concepts. Therefore, I need to make API calls for this set of prompts. 

 # API Calls 

 <api_calls> 
 <api>retrieve_information("Quantum mechanics principles")</api> 
 <api>get_instances("Quantum theory application")</api> 
 <api>ask_question("How to describe and analysis complex scientific concepts?")</api> 
 </api_calls> |
| **Default API Trajectory:** 
 # Analysis of Requirements 

 After analyzing the input prompts, I have identified the required background knowledge and broader capabilities. 

 # Competence Assessment 

 I have knowledge gaps and capability gaps. Therefore, I need to make API calls for this set of prompts. 

 # API Calls 

 <api_calls> 
 <api>ask_question("Explain the implications of the Einstein-Podolsky-Rosen paradox on quantum entanglement.")</api> 
 <api>ask_question("Describe the mathematical formulation of the wave function collapse in the Copenhagen interpretation.")</api> 
 <api>ask_question("Analyze the potential applications of quantum tunneling in developing next-generation transistors.")</api> 
 </api_calls> |

Table 4: An example of default and generated API trajectories for a set of observed instructions.

# D  DETAILS OF EVALUATION SETUPS

We first show the detailed implementation of our in-house evaluation and public benchmarks in Table 5. Then, we provide the corresponding task statistics of AlpacaEval 2.0, Arena-Hard, and MT-Bench in Table 6 and Table 7 by category and difficulty, respectively.

| Benchmark | # Task | # Ins. | Baseline Model | RM Judgment | GPT-4 Judgment |
|---|---|---|---|---|---|
| In-house Test Split | 1,000 | 97,000 | Original Policy Model | FsfairX-Llama-3-RM-v0.1 | GPT-4-1106-Preview |
| AlpacaEval 2.0 | 38 | 805 | GPT-4-1106-Preview | FsfairX-Llama-3-RM-v0.1 | GPT-4-1106-Preview |
| Arena-Hard | 34 | 500 | GPT-4-0314 | FsfairX-Llama-3-RM-v0.1 | GPT-4-1106-Preview |
| MT-Bench | 21 | 80 | GPT-4-0314 | FsfairX-Llama-3-RM-v0.1 | GPT-4-0125-Preview |

Table 5: Implementation details of evaluation on in-house test tasks and public benchmarks.

| Category | AlpacaEval 2.0 | | Arena-Hard | | MT-Bench | |
|---|---|---|---|---|---|---|
| | Number | Percentage | Number | Percentage | Number | Percentage |
| Advice Seeking | 79 | 9.8% | 12 | 2.4% | 4 | 5.0% |
| Brainstorming | 25 | 3.1% | 5 | 1.0% | 1 | 1.3% |
| Coding & Debugging | 42 | 5.2% | 204 | 40.8% | 7 | 8.8% |
| Creative Writing | 54 | 6.7% | 6 | 1.2% | 5 | 6.3% |
| Data Analysis | 7 | 0.9% | 66 | 13.2% | 3 | 3.8% |
| Editing | 73 | 9.1% | 19 | 3.8% | 5 | 6.3% |
| Information Seeking | 381 | 47.3% | 63 | 12.6% | 21 | 26.3% |
| Math | 40 | 5.0% | 56 | 11.2% | 17 | 21.3% |
| Planning | 61 | 7.6% | 45 | 9.0% | 3 | 3.8% |
| Reasoning | 27 | 3.4% | 17 | 3.4% | 11 | 13.8% |
| Role Playing | 14 | 1.7% | 4 | 0.8% | 3 | 3.8% |

Table 6: The task category statistics of AlpacaEval 2.0, Arena-Hard, and MT-Bench.

| Difficulty | AlpacaEval 2.0 | | Arena-Hard | | MT-Bench | |
|---|---|---|---|---|---|---|
| | Number | Percentage | Number | Percentage | Number | Percentage |
| Very Easy | 27 | 3.4% | 0 | 0.0% | 1 | 1.3% |
| Easy | 491 | 61.0% | 81 | 16.2% | 25 | 31.3% |
| Medium | 256 | 31.8% | 339 | 67.8% | 52 | 65.0% |
| Hard | 25 | 3.1% | 79 | 15.8% | 2 | 2.5% |
| Very Hard | 6 | 0.7% | 1 | 0.2% | 0 | 0.0% |

Table 7: The task difficulty statistics of AlpacaEval 2.0, Arena-Hard, and MT-Bench.

# E  COMPARISON BETWEEN ADS AND DATA CONSTRUCTION BASELINES

In this section, we expand our experimental evaluation by comparing the proposed ADS framework with several baseline methods for training data construction. These methods include: (1) **Prompting**, which constructs API trajectories through optimizer model prompting without fine-tuning; (2) **Retrieval Augmentation**, which employs both sparse and dense retrieval to identify relevant documents based on target task instructions, similar to the `Information Retrieval` API; (3) **Self-Instruct**, which utilizes the policy model to generate new instruction-response pairs for the target task, functioning analogously to the `Demonstration Generation` API; (4) **Rule-based QA**, which leverages an stronger LLM to answer each instruction in the target task, comparable to the `Question Answering` API. The empirical results presented in Table 8 indicate that ADS, incorporating all three APIs, significantly outperforms these baseline methods across both in-house test tasks and public benchmarks, and maintaining its simplification without human intervention.

| Methods | Qwen-2-7B-Instruct | | Gemma-2-9B-Instruct | |
|---|---|---|---|---|
| | In-house Test Tasks | Public Benchmarks | In-house Test Tasks | Public Benchmarks |
| Prompting | 36.7 | 27.2 | 67.0 | 33.7 |
| Retrieval Augmentation | 24.2 | 26.8 | 46.4 | 32.6 |
| Self-Instruct | 55.8 | 31.6 | 76.9 | 35.1 |
| Rule-based QA | 81.9 | 32.0 | 79.7 | 36.0 |
| ADS | **84.3** | **32.8** | **82.6** | **38.3** |

Table 8: Comparison between ADS and training data construction baseline methods.

# F  INFLUENCE OF THE INSTRUCTION-FOLLOWING DATASET

Since we adopt the instructions from Llama-3-Magpie-Air-3M-v0.1 (Xu et al., 2024) as the task instructions in our optimizer training process, we conduct a controlled experiment to ensure a fair comparison. Specifically, we fine-tune our base models using this dataset and evaluate their performance on public benchmarks before and after fine-tuning. As shown in Table 9, the fine-tuning process leads to a notable degradation in model performance. This decline can be attributed to the fact that our base models (Qwen-2-7B-instruct and Gemma-2-9-Instruct) have already undergone extensive fine-tuning with high-quality training data, whereas the Llama-3-Magpie-Air-3M-v0.1 dataset may contain relatively lower-quality data. These empirical findings substantiate that the performance improvements observed in our experiments stem from the effectiveness of the proposed ADS, rather than from the instruction-following dataset.

| Methods | Qwen-2-7B-Instruct | | | | Gemma-2-9B-Instruct | | | |
|---|---|---|---|---|---|---|---|---|
| | AlpacaEval 2.0 | Arena-Hard | MT-Bench | Average | AlpacaEval 2.0 | Arena-Hard | MT-Bench | Average |
| Base Model | **24.0** | **25.6** | **55.9** | **26.4** | **34.8** | **37.5** | **55.0** | **36.9** |
| Base Model w/ Magpie | 11.9 | 11.7 | 43.1 | 13.6 | 14.8 | 12.0 | 44.9 | 15.5 |

Table 9: Comparison between the base models before and after fine-tuning on the instruction-following dataset, where "w/ Magpie" denotes the base models after fine-tuning.

