# OpenReview forum: "Let Large Language Models Find the Data to Train Themselves"
_ICLR.cc/2025/Conference — ICLR 2025 Conference Withdrawn Submission_

### Official Review · Reviewer_fuzK · 2024-10-26

**Soundness:** 1
**Presentation:** 2
**Contribution:** 2
**Rating:** 3
**Confidence:** 4

**Summary:**

The paper presents the active data search (ADS) framework, allowing large language models to autonomously acquire training data without human supervision. ADS learns an "optimizer" via reinforcement learning to enable active data acquisition while minimizing costs. Experiments demonstrate that ADS leads to performance gains compared with the original model.

**Strengths:**

The method is well-motivated. Letting LLMs actively identify training data for themselves can potentially improve data efficiency and reduce human efforts in model training.

**Weaknesses:**

- **The true implementation of ADS is different from its description.** In Sec. 3.2, the authors claim that "_we **train** the original $\mathcal M^p$ on the tailored training dataset to update its knowledge and capacities_". This claim is very misleading because according to Appendix C, the so-called "training" is simply adding the collected data to the prompt. The authors refer to this as "in-context learning" (ICL). However, in [1], ICL is introduced as an _inference_ approach in contrast to _training_. Thus, I believe the main selling point in the title, "finding the data to _train_ the models", is a $\text{\color{red}significant overclaim}$, which can be very misleading to the community.

- **The paper does not compare ADS with any relevant baselines.** The authors include a few relevant papers in Sec. 2, e.g., [2,3]. The authors should compare the cost and quality of ADS with baselines to demonstrate its effectiveness.
- **I'm concerned about the efficiency of the method.** Algorithm 1 requires $\mathcal O(N|\mathcal A||\mathcal T|)$ evaluations of the policy model. No where in the paper do the author report the computation cost of the method. It's also not clear whether the method can scale up to more types of API calls and larger models in a truly practical scenario.
- Minor issues:
  - Line 50: This sentence is hard to read and grammatically  incorrect. Please consider the modification proposed by LLM: "We achieve the above automatic process through development of an optimizer that generates these APIs sequentially based on the target task to solve."
  - Line 58: `\citet` $\to$ `\citep`
  - Line 138: LLM-as-a judgment  $\to$ LLM-as-a judge

[1] Language Models are Few-Shot Learners

[2] Rlaif: Scaling reinforcement learning from human feedback with ai feedback.

[3] AutoDetect: Towards a Unified Framework for Automated Weakness Detection in Large Language Models

**Questions:**

Please refer to the **weakness** part.

---

> ### Author Response · Authors · 2024-11-28
> **Response to Reviewer fuzK**
>
> Thank you for your thoughtful review and valuable feedback. We appreciate your recognition of our work’s well-motivated. Below, we address your concerns in detail.
>
> > **Q1: Regarding the implementation of ICL for policy model updating.**
>
> **A1:** We appreciate the reviewer's insightful comment regarding the implementation of ICL for policy model updating. We would like to provide further clarification, the policy model requires frequent updates with a complexity of O(trajectory sampling number * target task number), resulting in approximately 40,000 updates per experiment. Given this computational intensity, implementing traditional fine-tuning approaches would be prohibitively expensive. Consequently, we adopted in-context learning as our primary methodology across all experimental conditions to maintain computational efficiency while preserving performance.
>
> Furthermore, extensive research has demonstrated that ICL can achieve comparable or even better effectiveness than traditional parameter fine-tuning [1][2][3][4][5]. We will include these analyses in the updated version.
>
> **References:**
>
> [1] Exploring the relationship between in-context learning and instruction tuning. arXiv preprint, 2023.
>
> [2] Few-shot Fine-tuning vs. In-context Learning: A Fair Comparison and Evaluation, ACL 2023 Findings, 2023.
>
> [3] Why Can GPT Learn In-Context? Language Models Implicitly Perform Gradient Descent as Meta-Optimizers. ACL 2023 Findings, 2023.
>
> [4] In-Context Learning with Long-Context Models: An In-Depth Exploration. arXiv preprint, 2024.
>
> [5] Many-Shot In-Context Learning. NeurIPS, 2024.
>
> > **Q2: Regarding the comparison between ADS and baseline methods.**
>
> **A2:** Since more than one reviewer raised this question, we have responded to it in the global rebuttal section (Q1) to save space.
>
> > **Q3: Regarding the efficiency of our method.**
>
> **A3:** We sincerely appreciate your insightful feedback regarding the efficiency of our method. Our approach is divided into two parts: optimizer refinement ("learn to learn") and policy optimization ("learn").
>
> Optimizer refinement requires frequent updates and evaluations of the policy model during the training process, resulting in a time complexity of O(trajectory sampling number * target task number). We acknowledge that the optimizer refinement process requires substantial initial overhead.
>
> However, once implemented, the subsequent costs for policy optimization become minimal. This automated approach serves to replace the traditional, labor-intensive process of manual training data collection and curation, which typically involves extensive trial and error, thereby significantly enhancing model adaptation efficiency for learning any tasks.
>
> Furthermore, as discussed in the limitations section, we acknowledge that the proposed ADS framework represents a conceptual prototype implementation of the automated data collection process. There remains substantial further investigation before practical implementation can be achieved. We will incorporate these analyses into the updated version of our work.
>
> > **Q4: Regarding the comparison between ADS and RLAIF and AutoDetect.**
>
> **A4:** We thank the reviewer for their insightful question regarding the comparison between ADS and AutoDetect. To clarify, the goal of our proposed ADS approach is to automatically collect valuable training data to enhance the performance of the target task, which distinguishes it from the aforementioned methods.
>
> The RLAIF approach primarily focuses on leveraging rewards from AI to replace human feedback in reinforcement learning. This method relies on existing training data and does not involve the collection of new data.
>
> In contrast, the AutoDetect method requires a powerful LLM, such as GPT-4, to identify weaknesses in a weaker policy model through iterative question generation and answer evaluation. This approach depends on GPT-4 for weakness identification and distills data from GPT-4.

---

> > ### Comment · Reviewer_fuzK · 2024-12-02
> >
> > The issue of overclaiming has not been resolved. The authors state that the computational cost is too high to perform actual training. If training cannot be conducted, then the claim of “training” should not be made.

---

> ### Author Response · Authors · 2024-12-02
>
> We acknowledge that the word "training" in our paper can be controversial. In-context learning is a new learning paradigm that significantly diverges from traditional machine learning approaches and primarily works with large language models. On one hand, a number of studies have studied its effectiveness, efficiency, and underlying mechanisms, particularly in comparison to fine-tuning, suggesting that it is a strong alternative to fine-tuning [1-5]. On the other hand, yes, it is usually categorized as an inference-time learning method.
>
> **To avoid future confusion, we can replace all related wordings of "training" with "teaching" in our revised manuscript.** Please note that the core contribution of this paper is the introduction of an automatic data collection and curation framework for targeted performance enhancement. Generally, the specific methods used to learn from the resulting data do not affect the overall framework. We can opt for fine-tuning when computational costs are more affordable.
>
> Thank you once again for your valuable suggestions. We greatly appreciate your feedback and would like to know if this helps clarify the contribution of our paper or if you have any further comments on this matter.

---

### Official Review · Reviewer_eS6a · 2024-10-27

**Soundness:** 3
**Presentation:** 3
**Contribution:** 3
**Rating:** 6
**Confidence:** 3

**Summary:**

The paper presents a self-improvement framework for LLMs that leverages external environments for data acquisition.

During training of the optimizer model, given a set of tasks and instructions for each task, the optimizer model learns (through RL) to generate optimized API call trajectories to acquire relevant external data, by measuring how the acquired data help improve performance of the policy model through finetuning or in-context learning on the data. The RL also takes cost control into account.

During test time, the optimizer model generates data acquisition calls for tasks and instructions unseen during training and the data acquired is used to improve the policy model for these tasks.

The experiments show that data acquisition significantly improves the policy model's performance on various tasks, including Alpaca Eval, Arena Hard, and MT bench, when relevant tasks and instructions are provided for training the optimizer model and with in-context learning of the policy model on the acquired data.

**Strengths:**

Originality: Self-improving LLMs via data acquisition is a novel direction, to the best of my knowledge.
Quality: The paper shows promising quality gains against the baselines.
Clarity: The framework and the complex process of training the optimizer model are clearly explained.
Significance: Self-improvement of LLMs is clearly an important direction. Acquiring data from an external environment will be critical.

**Weaknesses:**

Significance: The quality gains against a stronger baseline strategy that "utilizes the Question Answering API for each  observed instruction in the target task" are relatively minor.

Generalization: It's not clear how well the optimizer model generalizes across tasks. The experiments are set up so that the training tasks and instructions appear to be similar to those used during test time. The paper would be strong if it could demonstrate generalization across more distinct tasks.

**Questions:**

What are the important characteristics of the optimized trajectories that allow them to outperform the baseline trajectories?

What roles do "information retrieval", "demonstration generation", and "question answering" each play? What if we remove one of the APIs? How does their respective quality affect the final results?

The comparison against the "Question Answering for all instructions" strategy is only conducted for the in-house test set. Does it hold across benchmarks?

In Algorithm 1, "Split task instructions Q into observed set Qo and held-out set Qh" is in the iteration loop (t), does it mean that the split is different across iterations?

In section 4.3 "we employ the Self-Instruct approach to generate three new instructions for each task and use them as the observed instructions for that task". How much overlap is there between the generated instructions vs. the original ones?

In section 6.2, "In comparison to the approach that focuses solely on maximizing performance without considering costs, our method not only achieves a lower cost as expected but also demonstrates an improved win rate". It's not clear why cost control also improves quality. Is this due to randomness or some other factor?

---

> ### Author Response · Authors · 2024-11-28
> **Response to Reviewer eS6a (Part 1)**
>
> Thank you for your thoughtful review and valuable feedback. We appreciate your recognition of our work’s novelty and significance. Below, we address your concerns in detail.
>
> > **Q1: Regarding the intrinsic advantages of the API trajectory optimized by the optimizer model in ADS.**
>
> **A1:** We appreciate the reviewer’s insightful comment regarding the inherent advantages of API trajectories generated through our optimizer model. Within the ADS framework, the optimizer model is specifically designed to identify and curate specialized training datasets that enhance policy performance by generating appropriate textual API calls to interact with various resources and environments. Through iterative reinforcement learning, **the optimizer model is incentivized to detect and address performance weaknesses, thereby facilitating the progressive development of self-knowledge**. This process systematically increases the probability of generating optimal API trajectories, ultimately leading to continuous self-improvement of the system.
>
> > **Q2: Regarding the comparison between ADS and strong baseline strategies.**
>
> **A2:** Since more than one reviewer raised this question, we have responded to it in the global rebuttal section (Q1) to save space.
>
> > **Q3: Regarding the generalization of ADS.**
>
> **A3:** Thank you for the insightful question. We would like to clarify that the primary objective of our ADS framework is to develop a generalized optimizer model capable of addressing a wide spectrum of target tasks. To achieve this, we have meticulously compiled a diverse dataset comprising approximately 10,000 target tasks for optimizer training and validation.
>
> Additionally, in our experiments, despite the evaluation of 1,000 in-house target tasks, we have also validated our optimizer model's generalizability on three established public benchmarks: AlpacaEval 2.0, Arena-Hard, and MT-Bench. These benchmarks span multiple domains and task types, and importantly, are entirely independent of our training dataset, thereby providing robust evidence of our model's generalization applicability.
>
> > **Q4: Regarding the function and importance of the three APIs in ADS.**
>
> **A4:** Thank you for raising this important point. As shown in Section 4.1, these three APIs are designed to facilitate the acquisition, utilization, and enhancement of knowledge, respectively. The Information Retrieval API employs both sparse and dense retrieval to retrieve relevant documents from external knowledge databases such as Wikipedia, thereby supporting knowledge acquisition, analogous to the pre-training stage of LLMs. The Demonstration Generation API utilizes the policy model to generate appropriate exemplar instruction-response pairs, tailored to various knowledge application scenarios, reminiscent of the alignment stages of LLMs. The Question Answering API resorts to the wisdom of human experts, mimicking how humans learn from each other.
>
> We also demonstrate the ablation experimental results in the table below. We can observe that **the proposed ADS framework with all of these three APIs demonstrates the best performance**.
>
> | Methods                      | Qwen-2-7B-Instruct  |                   | Gemma-2-9b-Instruct |                   |
> |------------------------------|---------------------|-------------------|---------------------|-------------------|
> |                              | **In-House Test Tasks** | **Public Benchmarks** | **In-House Test Tasks** | **Public Benchmarks** |
> | Information Retrieval API    |         24.2        |        26.8       |         46.4        |        32.6       |
> | Demonstration Generation API |         55.8        |        31.6       |         76.9        |        35.1       |
> | Question Answering API       |         81.9        |        32.0       |         79.7        |        36.0       |
> | ADS (All APIs)               |         **84.3**        |        **32.8**       |         **82.6**        |        **38.3**       |
>
> > **Q5: Regarding the splitting of observed and held-out instructions.**
>
> **A5:** Thank you for raising this thoughtful question. As discussed in Section 4.2 (line 226-233), we divide the instructions for each target task into two distinct sets: one set of observed instructions for trajectory generation, and another set of held-out instructions for performance evaluation. This splitting serves a crucial purpose: it enables us to verify that the optimizer is effectively learning to enhance task-level performance (i.e., acquiring valuable training data that improves overall task capability) rather than merely optimizing for instance-level solutions (i.e., developing specific solutions for given instructions). This distinction is important because superior performance on observed instructions does not necessarily generalize to improved performance on held-out instructions, potentially indicating overfitting rather than effective learning.

---

> ### Author Response · Authors · 2024-11-28
> **Response to Reviewer eS6a (Part 2)**
>
> > **Q6: Regarding the overlap between generated instructions and original instructions.**
>
> **A6:** Thank you for raising this important point. In our methodology for generating new task-specific instructions (which serve as observed instructions), we follow the implementation in Alpaca [1] to ensure instruction diversity. Specifically, we prompted the LLM to generate instructions that were different from the original ones. To maintain distinctiveness, we employed a filtering mechanism whereby any generated instructions with a Rouge-L similarity score exceeding 0.7 when compared to the original instructions were eliminated. We will incorporate these details in the revised version.
>
> **References:**
>
> [1] Stanford Alpaca: An Instruction-following LLaMA model. GitHub repository, 2023.
>
> > **Q7: Regarding the reasons for performance improvements achieved by cost control.**
>
> **A7:** This is an insightful observation. The cost-control mechanism, by selecting responses from among those with top-tier rewards while prioritizing lower costs, inherently promotes greater response diversity compared to pure reward maximization approaches. This enhanced diversity potentially yields two significant benefits: first, it contributes to more generalized performance improvements across various tasks, and second, it helps mitigate reward over-fitting issues. These additional discussions will be incorporated into our revised manuscript.

---

> > ### Comment · Reviewer_eS6a · 2024-12-03
> >
> > Thanks for the additional ablation study, which confirms the reviewers' earlier concern that the gains mostly come from the QA API. I will keep the current score.

---

> ### Author Response · Authors · 2024-12-03
>
> We sincerely appreciate the reviewer's insightful comment regarding the utilization of the QA API within the ADS framework.
>
> We would like to clarify that our proposed ADS framework does not overly rely on the QA API; for instance, on 1,000 in-house tasks using Gemini-2-9B-Instruct, the number of QA API calls was reduced by 59.6% compared to the ablation baseline that solely depends on the QA API. The proportion of QA usage is only about 44.2% among all three APIs. It is important to note that the QA API necessitates a strong external LLM to serve as a proxy for human annotation, making it the most cost-intensive among the other APIs. The ablation study actually shows that our ADS framework achieves larger performance gains at reduced operational costs, particularly with much fewer QA API calls.
>
> Thank you once again for your valuable suggestions. We greatly appreciate your feedback and hope that above information helps address your concern regarding the effect of the QA API.

---

### Official Review · Reviewer_6jxd · 2024-11-02

**Soundness:** 3
**Presentation:** 3
**Contribution:** 3
**Rating:** 6
**Confidence:** 2

**Summary:**

This paper introduces an automated framework called ACTIVE DATA SEARCH (ADS), designed to enable large language models (LLMs) to autonomously discover and acquire valuable training data for self-improvement without the need for human supervision.
The authors propose using reinforcement feedback signals to guide the models in selecting optimal data, rewarding performance gains while penalizing computational overhead.
The framework is validated through extensive experiments on 1,000 diverse in-house test tasks and three public benchmarks, demonstrating significant performance improvements.

**Strengths:**

(1) The ADS framework is innovative as it leverages LLMs to autonomously enhance their training data, reducing the need for costly human intervention and addressing scalability issues. The use of reinforcement feedback signals to balance performance improvement and computational efficiency is a practical and smart approach.
(2) Empirical results are robust, including 1,000 in-house test tasks and three public benchmarks, showing clear performance gains and generalization capabilities. The inclusion of detailed implementation protocols, such as API designs for data acquisition, is valuable for reproducibility.
(3) The paper presents a method for iterative refinement, demonstrating consistent performance improvements with the optimizer refinement through reinforcement learning.

**Weaknesses:**

(1) The paper would benefit from more comprehensive ablation studies. Specifically, it would be insightful to understand the individual contributions of each proposed API and the iterative refinement process. For example, what is the impact of excluding one of the APIs, or how does the system perform without the iterative refinement?
(2) The paper does not sufficiently discuss the potential limitations and failure cases of the ADS framework. Identifying scenarios where the framework might not perform well or discussing any observed limitations during the experiments would provide a more balanced view.
(3) Comparisons with more diverse sets of baselines, especially in terms of data acquisition strategies, would strengthen the validation of the effectiveness of the proposed framework.
(4) The exposition around reinforcement learning strategies could be expanded to aid comprehension, especially for readers less familiar with advanced reinforcement learning techniques.

**Questions:**

(1) Could you perform additional ablation studies to analyze the impact of each data acquisition API within the framework? Understanding the individual contributions of each API would highlight their importance and the overall robustness of the ADS framework.
(2) Were there any particular scenarios or tasks where the ADS framework did not perform as expected? If so, what were these failure cases, and how were they addressed or mitigated in the study?
(3) Could you provide further comparisons with alternative data acquisition methods beyond the ones mentioned in the paper?
(4) How sensitive is the ADS framework to different policy model architectures and sizes?

---

> ### Author Response · Authors · 2024-11-28
> **Response to Reviewer 6jxd (Part 1)**
>
> Thank you for your thoughtful review and valuable feedback. We appreciate your recognition of our work’s innovation and robustness. Below, we address your concerns in detail.
>
> > **Q1: Regarding the comparison between ADS and baseline methods.**
>
> **A1:** Since more than one reviewer raised this question, we have responded to it in the global rebuttal section (Q1) to save space.
>
> > **Q2: Regarding the contribution of individual API and iterative refinement.**
>
> **A2:** Thank you for raising this important point. As shown in Section 4.1 (line 226-233), these three APIs are designed to facilitate the acquisition, utilization, and enhancement of knowledge, respectively. The Information Retrieval API employs both sparse and dense retrieval to retrieve relevant documents from external knowledge databases such as Wikipedia, thereby supporting knowledge acquisition, analogous to the pre-training stage of LLMs. The Demonstration Generation API utilizes the policy model to generate appropriate exemplar instruction-response pairs, tailored to various knowledge application scenarios, reminiscent of the alignment stages of LLMs. The Question Answering API resorts to the wisdom of human experts, mimicking how humans learn from each other.
> We also demonstrate the ablation experimental results in the table below. We can observe that **the proposed ADS framework with all of these three APIs demonstrates the best performance**.
>
> | Methods                      | Qwen-2-7B-Instruct  |                   | Gemma-2-9b-Instruct |                   |
> |------------------------------|---------------------|-------------------|---------------------|-------------------|
> |                              | **In-House Test Tasks** | **Public Benchmarks** | **In-House Test Tasks** | **Public Benchmarks** |
> | Information Retrieval API    |         24.2        |        26.8       |         46.4        |        32.6       |
> | Demonstration Generation API |         55.8        |        31.6       |         76.9        |        35.1       |
> | Question Answering API       |         81.9        |        32.0       |         79.7        |        36.0       |
> | ADS (All APIs)               |         **84.3**        |        **32.8**       |         **82.6**        |        **38.3**       |
>
> To illustrate the effectiveness of the iterative refinement process in ADS, our original submission conducted a series of experiments and demonstrated the iterative training results (iteration 0-3) in Figure 2 and Table 2. Our findings indicate that **iterative ADS boosts consistent performance improvements for both in-house test tasks and generalized public benchmarks**.
>
> > **Q3: Regarding the analysis of tasks that ADS does not perform as expected.**
>
> **A3:** Thank you for raising this important point. In Figure 3, we have shown that for categories like editing, creative writing, and debugging, our ADS only has slight improvements or maintains comparable to the baseline.  This limited enhancement can be potentially attributed to the inherent nature of these tasks, which primarily involve format and style rewriting, as well as fragment modifications, presenting inherent challenges for optimization through in-context learning from acquired training data. These additional analyses will be incorporated into the updated version of the paper.
>
> > **Q4: Regarding the details of reinforcement learning.**
>
> **A4:** We sincerely appreciate your insightful feedback regarding the details of reinforcement learning. To facilitate the optimizer’s decision-making process in obtaining optimal training data, we propose a reinforcement learning strategy for optimizer training. This strategy leverages feedback reward signals from the policy to concurrently maximize task performance and minimize computational costs.
> As described in Section 4.4 (line 307-311), the reinforcement learning process comprises two primary phases. Initially, we implement a warm-up rejection sampling procedure, which fine-tunes the optimizer model on trajectories that demonstrate the highest reward among multiple sampled responses. Subsequently, we engage in an iterative process to update the optimizer model through direct preference optimization. This process involves selecting a "chosen" trajectory (exhibiting the highest reward) against a "rejected" trajectory (displaying the lowest reward). The objective of this approach is to enable the optimizer model to amplify the gap between these high-quality and low-quality trajectories, therefore increasing the probabilities of generating valuable API trajectories.

---

> ### Author Response · Authors · 2024-11-28
> **Response to Reviewer 6jxd (Part 2)**
>
> > **Q5: Regarding the implementation of ADS to different policy models.**
>
> **A5:** Thank you for this valuable suggestion. In our experiments, we have explored two different policy models, including Qwen-2-7B-Instruct and Gemma-2-9B-Instruct, both of which are famous open-source LLMs in the research community. As illustrated in Figure 2 and Table 2, our proposed ADS framework, when integrated with these diverse policy models, demonstrates significant performance enhancements across a wide range of in-house and public test tasks.

---

> ### Comment · Reviewer_6jxd · 2024-11-28
> **Thank you to the authors**
>
> Thank you for your replies. Most of my concerns have been addressed.

---

> > ### Author Response · Authors · 2024-11-28
> > **Thanks**
> >
> > Thanks for your positive feedback and for the constructive comments that are pivotal to improve our work.

---

### Official Review · Reviewer_EL9t · 2024-11-03

**Soundness:** 2
**Presentation:** 3
**Contribution:** 2
**Rating:** 3
**Confidence:** 4

**Summary:**

The paper introduces the Active Data Search (ADS) framework, which enables large language models (LLMs) to autonomously identify and acquire training data to improve their own performance with minimal human intervention. The authors propose an optimizer model that generates API calls for data collection from various external sources, such as web search engines, AI assistants, and annotation services. This framework leverages reinforcement learning to refine the optimizer, balancing task performance enhancement with computational cost. Experimental results, using models like Qwen-2-7B-Instruct and Gemma-2-9B-Instruct, show substantial performance gains in in-house tasks and public benchmarks, demonstrating ADS’s effectiveness in enabling smaller models to achieve results comparable to larger ones.

**Strengths:**

The paper presents the ADS framework, which automates the identification and acquisition of training data, enhancing the self-improvement capabilities of LLMs. This direction holds promise for reducing reliance on human intervention, making model development more efficient and scalable.

**Weaknesses:**

While the paper offers an innovative approach, it has several areas that could be strengthened. The lack of comparison with diverse baselines limits the assessment of the ADS framework's true efficacy. Specifically, comparisons with
(a) LLMs using basic RAG for retrieval,
(b) untrained LLMs collecting data for policy model inference,
(c) training the optimizer model with simple rule-based or manually collected data,
(d) other naive automated data collection techniques would provide deeper insights into the relative advantages of ADS.
Including these baselines would enhance the evaluation's comprehensiveness and validate the framework's practical significance against existing methods.

**Questions:**

1. Relationship with RAG: The paper lacks a thorough discussion on how the proposed ADS framework relates to or differs from Retrieval-Augmented Generation (RAG). Given that both approaches involve external data retrieval, it would be beneficial to address whether ADS extends, overlaps, or diverges from RAG in terms of methodology, application, or objectives. This comparison would help position the contribution within the broader context of retrieval-based techniques and clarify its novelty.

2. Inconsistent Use of Llama in Experiments: The absence of Llama-based experiments in Figure 2, despite its inclusion as a comparison point in Table 2, raises questions about consistency. It would be helpful for the authors to explain why Llama models were not tested under the same conditions as other models, or why their results were only included in specific sections. This would ensure a clearer understanding of the comparative analysis and model choice rationale.

---

> ### Author Response · Authors · 2024-11-28
> **Response to Reviewer EL9t**
>
> Thank you for your thoughtful review and valuable feedback. We appreciate your recognition of our work’s promisingness and effectiveness. Below, we address your concerns in detail.
>
> > **Q1: Regarding the comparison between ADS and baseline methods.**
>
> **A1:** Since more than one reviewer raised this question, we have responded to it in the global rebuttal section (Q1) to save space.
>
> > **Q2: Regarding the difference between ADS and RAG.**
>
> **A2:** Thank you for your insightful question. The distinctions between ADS and RAG can be summarized across three key dimensions:
>
> 1. **Motivation:** ADS primarily aims to identify valuable training data for a specific task through environmental interactions via various APIs (including but not limited to Information Retrieval), ultimately enhancing overall task performance. In contrast, RAG focuses on retrieving relevant information for individual queries, with the primary objective of improving response quality for specific queries rather than optimizing global task performance.
> 2. **Implementation:** The fundamental component of ADS is the optimizer model, which dynamically generates appropriate API calls and API parameters at the task level. This differs significantly from traditional RAG, where its retrieval process solely and directly matches the given instructions to texts in a database.
> 3. **Framework:** ADS incorporates more comprehensive APIs beyond Information Retrieval, including Demonstration Generation and Question Answering. This integration of diverse APIs results in a more complicated and effective framework compared to RAG.
>
> > **Q3: Regarding the use of the Llama model in our experiments.**
>
> **A3:** We sincerely appreciate your insightful feedback regarding the use of the Llama model in our experiments. To address your questions, we would like to clarify that, **since our training data is generated by Llama-3-8B-Instruct, we exclude it from our choices of policy models to avoid any potential biases and ensure a fair comparison**. This explanation is also mentioned in Section 4.4 (line 323) of the current manuscript.
> Regarding the Question Answering API, which is designed to generate responses to given questions and is independent of the training data, we opt for the strong Llama-3.1-70B-Instruct model as a proxy for human annotation in practice for efficiency and reproducibility. It is worth noting that alternative LLM such as GPT-4 or Claude-3.5 could also serve as the strong model.

---

### Official Review · Reviewer_CBK9 · 2024-11-11

**Soundness:** 2
**Presentation:** 3
**Contribution:** 2
**Rating:** 3
**Confidence:** 5

**Summary:**

Currently, LLM developers manually analyze model errors and engineer training datasets (be it through human labels, or synthetically) to enhance model performance across pre-training, instruction tuning, and preference learning stages. This approach is costly, error-prone, heavily reliant on developer expertise, and lacks scalability. To address these limitations, this paper introduces the **Active Data Search (ADS)** framework, where an LLM assesses its own strengths and weaknesses for a given task and autonomously collects relevant training data to improve itself.

At the core of ADS is an LLM-driven optimizer that, when given a target task (grounded by a few guiding questions), evaluates the capabilities of the primary LLM (referred to as the policy LLM) for the task and dynamically calls different data retrieval and generation APIs to collect relevant data for improvement. This data can then be used to fine-tune the policy LLM for the target task or enhance it via in-context prompt augmentation (as implemented in this paper). Since LLMs typically are not designed for this autonomous data collection task, ADS first trains this optimizer offline with reinforcement learning to balance the reward and cost associated with obtaining new data.

The paper employs three simple data collection APIs: (a) information retrieval from Wikipedia, (b) demonstration generation powered by an LLM, and (c) question-answering using an LLM. Trained on the MagPie dataset, the optimizer effectively learns to use these APIs to acquire relevant data, leading to improved performance on an internal test set and on 3 additional public benchmarks (AlpacaEval, MT-Bench, and Arena-Hard) across two different LLMs, at times matching the performance of larger LLMs.

**Strengths:**

1. The paper tackles the important challenge of gathering optimal data to enhance LLM performance for a given task.
2. The proposed framing, where LLMs become self-aware of their strengths and weaknesses to identify optimal data for self-improvement, is compelling. This framing has significant implications, not only for advancing the next generation of state-of-the-art LLMs but also for democratizing AI by empowering non-experts to tailor LLMs to their specific needs.
3. The proposed ADS method of framing diverse data collection techniques as API/Tool calls and training an optimizer using DPO to dynamically select the most effective data collection API calls for a given task, is innovative and powerful.
4. The paper’s related work section is well-structured, effectively positioning the authors' contributions within the context of prior research.
5. Experimental evaluations demonstrate substantial improvements over the baseline Qwen-2-7b-instruct model across several public benchmarks.

**Weaknesses:**

The core weaknesses of this paper can be organized along the following broad themes:

---
### Insufficient Experimental Validation
---
The paper lacks comparisons with key baselines and prior work, making it challenging to fully assess the contributions of this approach. Specifically:

1. **Dynamic API Selection Validation**: While the individual data collection APIs—Information Retrieval, Demonstration Generation, and Question Generation—are well-studied in prior work as Retrieval-Augmented Generation and its variants [1], Self-Instruct [2], and behavior cloning/distillation from teacher LLMs respectively, this work’s primary novelty lies in an optimizer that dynamically selects among these APIs. However, the paper does not compare this dynamic selection against these established methods, leaving unclear whether it is indeed critical for improved performance. To clarify, a comparison is recommended in the following settings:

    - *Inference-only Comparison*: Evaluate ADS against comparable-sized synthetic datasets generated by prior methods, as well as a larger dataset from these methods, given that they do not require additional compute for optimizer training.
    - *Optimizer Training with Single APIs*: Train ADS-style optimizers with individual APIs (e.g., Information Retrieval only) on the MagPie dataset, to assess the relative impact of each API within ADS.

&nbsp;

2. **MagPie Fine-Tuned Baseline**: Unlike ADS, which is trained on the MagPie dataset, baseline LLMs are not. Since fine-tuning on MagPie has been shown to enhance performance on target benchmarks, this gives ADS an advantage over baseline models. A more balanced comparison would involve fine-tuning baseline LLMs on MagPie to assess if ADS's optimizer training and synthetic data generation offer distinct advantages over straightforward fine-tuning.

&nbsp;

---
### Scalability Concerns & Scaling Experiments
---
3. **Generated Dataset Size and Limitations**: The paper studies the proposed method in an in-context policy improvement setting, where synthetic data is added directly to the prompt. This restrictive setting limits the amount of synthetic data that can be utilized, as it must fit within the model's maximum sequence length. Discussion on scenarios where generated data exceeds this length and experiments studying the impact of synthetic data scaling on the method’s effectiveness would strengthen the paper.

4. **Input Task-Specific Dataset Size**: The method currently relies on only five labeled instances from the target task to generate synthetic data, whereas real-world applications may allow access to more extensive task-specific data. Including a discussion on method scalability to larger datasets (e.g., prompt with all task data at once or using sampling and multi-prompting) and experiments exploring the impact of task-specific data scaling would be beneficial.

5. **Method Extension and Efficacy for Fine-Tuning**: Since the method is limited to in-context policy improvements, its effectiveness in policy fine-tuning contexts remains untested. Additionally, the feasibility of using the current optimizer training algorithm for fine-tuning is uncertain, as retraining policies to obtain reward for each data generation step could be prohibitively costly. Discussion of this limitation and experiments in fine-tuning scenarios could further clarify the method's general applicability.

&nbsp;

---
### Method Robustness Concerns
---
6. **Optimizer Generality**: It is unclear if the optimizer learns generalizable insights into model strengths and weaknesses or if its effectiveness is limited to specific tasks. Current training and testing datasets are closely related (e.g., MagPie -> AlpacaEval, etc.). Testing on unrelated benchmarks, like Big-Bench-Hard, GSM8k, or MMLU, would offer insights into the optimizer’s generality.
7. **Dependence on MagPie Dataset**: The extent to which the optimizer’s performance depends on the choice of MagPie as the training dataset is unclear. Evaluating its performance when trained on different datasets would provide evidence of ADS's broader applicability.
8. **Reliance on Task Clustering**: ADS clusters similar instances into tasks to generate synthetic data during both training and testing. This clustering may not always be feasible and could require manual intervention. Without this careful clustering, retrieving the relevant synthetic data for a test instance could become a challenge, raising questions about the method's performance if instances are grouped differently, such as randomly, during data generation.

&nbsp;

---
### Limited analysis
---
Several important areas lack critical analysis in the paper:

9. **Generated Synthetic Data**: The paper does not analyze the characteristics or quality of the synthetic data generated. A manual review of the generated data could provide valuable insights into its role in improved performance.

10. **API Call Patterns**: There is no examination of the API calls made by the optimizer at test time. Analyzing the frequency and distribution of these calls could clarify the optimizer's behavior and the relative importance of each data source (API).

11. **Weakness Reflection Rationales**: Although the paper claims that the optimizer uses LLM-driven reflections to assess model strengths and weaknesses, it lacks a qualitative or manual analysis of these reflections. This omission makes it challenging to gauge the accuracy and depth of these self-assessments.

&nbsp;

References:

[1] Yu, Wenhao, et al. "Generate rather than retrieve: Large language models are strong context generators." arXiv preprint arXiv:2209.10063 (2022).

[2] Wang, Yizhong et al. “Self-Instruct: Aligning Language Models with Self-Generated Instructions.” Annual Meeting of the Association for Computational Linguistics (2022).

**Questions:**

Please refer to the *Weakness* section above for recommendations on experiments and analyses that could further strengthen the paper.

Additional questions and suggestions **that do not impact the score**:

1. How are preference pairs constructed from sampled API trajectories for rejection sampling and DPO? Is a single preference pair created from the highest and lowest reward trajectories?
2. Is the cost of API calls considered only when selecting an efficient trajectory from a trajectory tier (group) or do you also add a cost term to the overall reward?
3. Are all prompts (individual APIs, optimizer, evaluation prompts) zero-shot? If not, how many demonstrations are provided for each?
4. Consider expanding the discussion on other data collection techniques (APIs) that could be integrated into the framework.
5. The "internal test-set" is not described in the paper. Without additional details about this internal test set, it is difficult to assess the quality of the improvements. Is it just the test partition of the MagPie dataset?

---

> ### Author Response · Authors · 2024-11-28
> **Response to Reviewer CBK9 (Part 1)**
>
> Thank you for your thoughtful review and valuable feedback. We appreciate your recognition of our work’s innovation and significance. Below, we address your concerns in detail.
>
> > **Q1: Regarding the comparison between ADS and baseline methods.**
>
> **A1:** Since more than one reviewer raised this question, we have responded to it in the global rebuttal section (Q1) to save space.
>
> > **Q2: Regarding the influence of the Magpie dataset.**
>
> **A2:** Thank you for your valuable suggestion. Firstly, we opted for the Magpie dataset due to its comprehensive coverage of diverse alignment tasks, encompassing various domains, difficulty levels, and intents. This aligns with the objective of our ADS framework, which aims to identify suitable data for training policy models for each specific task. Furthermore, to ensure a fair comparison, we fine-tuned our base LLMs using the Magpie dataset and conducted evaluations on public benchmarks.
>
> | Methods            | Qwen-2-7B-Instruct |            |          |         | Gemma-2-9B-Instruct |            |          |         |
> |--------------------|--------------------|------------|----------|---------|---------------------|------------|----------|---------|
> |                    |   **AlpacaEval 2.0**   | **Arena-Hard** | **MT-Bench** | **Average** |    **AlpacaEval 2.0**   | **Arena-Hard** | **MT-Bench** | **Average** |
> | Base LLM           |        **24.0**        |    **25.6**    |   **55.9**   |   **26.4**  |         **34.8**        |    **37.5**    |   **55.0**   |   **36.9**  |
> | Base LLM w/ Magpie |        11.9        |    11.7    |   43.1   |   13.6  |         14.8        |    12.0    |   44.9   |   15.5  |
>
> Our empirical results indicate that **fine-tuning the LLM with Magpie data significantly decreased performance**. The reason is that the base LLMs (Qwen-2-7B-instruct and Gemma-2-9-Instruct) have already undergone extensive fine-tuning with high-quality training data. Compared to that, the data quality of Magpie may be relatively low. These findings confirm that the improvements observed in our experiments are not attributed to the Magpie dataset, indicating the effectiveness of the proposed ADS method. We will incorporate these findings into the revised manuscript.
>
> > **Q3: Regarding the influence of generated data length on ADS.**
>
> **A3:** We appreciate your insightful feedback and would like to offer some clarification. It is important to note that when employing in-context learning for policy updating, the scale of generated data is inherently constrained by the model's context window capacity. However, state-of-the-art LLMs typically operate with context windows of approximately 32,000 tokens. Our empirical observations indicate that this scale of training data is sufficient for the effective optimization of a given target task.
>
> > **Q4: Regarding the analysis of input task-specific dataset size to the optimizer model.**
>
> **A4:** We appreciate your insightful question. In our implementation, we utilize the task-specific examples as a representative for the target task. Since successful task completion requires various fundamental capabilities and skills, we speculate that more observed examples could represent a broader spectrum of task scenarios, therefore achieving better performance. Currently, we are conducting experiments to scale the example number, and we will update the results once the experiment is concluded.
>
> > **Q5: Regarding the extension from ICL to fine-tuning.**
>
> **A5:** We appreciate the reviewer's insightful comment regarding the implementation of ICL for policy model updating. We would like to provide further clarification, the policy model requires frequent updates with a complexity of O(trajectory sampling number * target task number), resulting in approximately 40,000 updates per experiment. Given this computational intensity, implementing traditional fine-tuning approaches would be prohibitively expensive. Consequently, we adopted in-context learning as our primary methodology across all experimental conditions to maintain computational efficiency while preserving performance.
> Furthermore, extensive research has demonstrated that **ICL can achieve comparable or even better effectiveness than traditional parameter fine-tuning** [1][2][3][4][5]. We will include these analyses in the updated version.
>
> **References:**
>
> [1] Exploring the relationship between in-context learning and instruction tuning. arXiv preprint, 2023.
>
> [2] Few-shot Fine-tuning vs. In-context Learning: A Fair Comparison and Evaluation, ACL 2023 Findings, 2023.
>
> [3] Why Can GPT Learn In-Context? Language Models Implicitly Perform Gradient Descent as Meta-Optimizers. ACL 2023 Findings, 2023.
>
> [4] In-Context Learning with Long-Context Models: An In-Depth Exploration. arXiv preprint, 2024.
>
> [5] Many-Shot In-Context Learning. NeurIPS, 2024.

---

> ### Author Response · Authors · 2024-11-28
> **Response to Reviewer CBK9 (Part 2)**
>
> > **Q6: Regarding the generalization of ADS.**
>
> **A6:** Thank you for the insightful question. We would like to clarify that the primary objective of our ADS framework is to develop a generalized optimizer model capable of addressing a wide spectrum of target tasks. To achieve this, we have meticulously compiled a diverse dataset comprising approximately 10,000 target tasks for optimizer training and validation.
> Additionally, in our experiments, despite the evaluation of 1,000 in-house target tasks, we have also validated our optimizer model's generalizability on three established public benchmarks: AlpacaEval 2.0, Arena-Hard, and MT-Bench. These benchmarks span multiple domains and task types, and importantly, are entirely independent of our training dataset, thereby providing robust evidence of our model's generalization applicability.
>
> > **Q7: Regarding the requirements of clustered tasks.**
>
> **A7:** We appreciate the reviewer’s insightful comment regarding the requirements of clustered tasks. Our proposed approach primarily aims to enable LLMs to identify valuable training data that can enhance their performance on specific target tasks. In our experiments, we prompt LLMs to cluster a large instruction dataset into splits of specific tasks, ensuring the reproducibility of our research. We believe that in practical applications, users can simply formulate a small number of exemplar instructions according to their needs.
>
> > **Q8: Regarding the case studies of ADS.**
>
> **A8:** Thank you for raising this important question. In our original submission, we present case studies in Table 4 (Appendix C). We can observe that our optimizer model, when provided with a few task-specific instructions, is capable of executing a three-step process. First, it generates a comprehensive analysis of the fundamental requirements to solve the target task. Subsequently, it produces self-reflective evaluations to identify potential limitations. Finally, it develops corresponding API trajectories to construct suitable training data, thereby enhancing task-specific capabilities.
>
> > **Q9: Regarding the construction of API trajectories in rejection sampling and direct preference optimization.**
>
> **A9:** Thank you for raising this important question. As discussed in Section 4.4 (line 307-311), for reward maximization optimization, the construction of API trajectories is as follows:
>
> 1. In rejection sampling, the chosen trajectory is the one with the maximum reward from multiple sampled trajectories.
> 2. In direct preference optimization, we select the paired chosen and rejected trajectories with the maximum and minimum rewards, respectively.
>
> For the cost-control mechanism, we introduce a cost tier parameter τ ∈ [0, 1] to control the trade-off between rewards and costs. Trajectories within the top-tier rewards ranging [(1−τ )Rmax+τRmin, Rmax] are considered to have similar performance. From this subset, we select the trajectory with the lowest cost as the chosen trajectory. Conversely, for the reject trajectory, we select the one with the highest cost within [Rmin,(1 − τ )Rmin + τRmax]. We will make the above clearer in the next version.
>
> > **Q10: Regarding the cost of API calls considered only when selecting trajectories.**
>
> **A10:** Yes, you are right. The API call costs are exclusively considered during the selection of trajectories from a given reward-tier trajectory group. Specifically, for rejection sampling, we select the chosen trajectory with the minimal cost among those in the highest reward tier. Regarding direct preference optimization, we select the lowest-cost trajectories from the top reward tier for chosen trajectories, and highest-cost trajectories from the bottom reward tier for rejected trajectories.
>
> > **Q11: Regarding the details of the prompts used in ADS.**
>
> **A11:** We would like to note that all these prompts are detailed in Appendix A of our original submission. For individual API, we utilize zero-shot prompting. For the optimizer model's API trajectory generation, we implement a two-shot prompting, incorporating examples both with and without API calls. Regarding the evaluation prompts, we maintain consistency with the official implementations by using zero-shot prompts throughout the assessment process.
>
> > **Q12: Regarding the details of the internal test set.**
>
> **A12:** Thank you for raising this important point. Our internal test set comprises 1,000 target tasks, which were constructed from the Magpie dataset. To ensure robust evaluation, for the test set, we expanded each task's initial five instructions to 100 through the Self-Instruct method, where three instructions serve as observed samples while the remaining 97 are maintained as held-out examples. We have provided a detailed description of the test set construction method in Section 4.2 (lines 254-262), with comprehensive statistical analyses in Table 3 (Appendix B).

---

> > ### Comment · Reviewer_CBK9 · 2024-12-03
> > **Response to the Author Rebuttal**
> >
> > Thank you for the additional experiments and detailed rebuttal. While I appreciate the effort put into addressing my concerns, I am inclined to maintain my current score, as some key issues remain unresolved:
> >
> > 1. The method remains intractable for "training" better policies. Furthermore, its scalability with respect to the number of input demonstrations and the size of the output synthetic dataset in its in-context policy improvement setting is still unknown. No additional experiments were provided to evaluate the method's performance when these parameters are varied, which raises concerns about its practicality and broader applicability.
> >
> > 2. Analysis of Synthetic Data: The paper still lacks a thorough qualitative and quantitative analysis of the generated synthetic data, API trajectories and distributions, and the "weakness" rationales produced by the model. The single illustrative example in Appendix C is insufficient to evaluate these aspects comprehensively.
> >
> > 3. Robustness Concerns: Robustness remains an open question, as there are no experiments evaluating how task clustering quality (e.g., carefully grouped vs. randomly grouped tasks) affects performance, or assessing the optimizer's efficacy when trained on datasets other than MagPie.
> >
> > 4. Baseline Details and Comparisons: While the authors included new experiments with a MagPie fine-tuned baseline (Appendix F) and ablation studies for different APIs (Appendix E), critical details about these baselines are missing. For example, it is unclear whether the synthetic dataset sizes generated by various methods are comparable, or how the MagPie fine-tuned baseline was trained (e.g., model selection process, validation task performance, whether LoRA fine-tuning or full fine-tuning was used, etc.). These details are essential for interpreting the results, particularly given the surprising drop in performance for the MagPie-trained baseline.

---

### Author Response · Authors · 2024-11-28
**General Response to all Reviewers**

We sincerely appreciate the time and effort all the reviewers put into reviewing our paper. In the subsequent points, we will carefully address the common concerns of our paper.

> **Q1: Regarding the comparison between ADS and baseline methods.**

**A1:** We appreciate the reviewer's concern regarding the comparison between ADS and baseline methods.  To clarify, our original submission included two types of baselines in our experiments. Specifically, we evaluated

1. **Prompting**, which constructs API trajectories through optimizer model prompting without fine-tuning (the prompting method in Figure 2 and Table 2).
2. **Rule-based QA**, which utilizes the Question Answering API for each observed instruction in the target task to construct the corresponding API trajectory (the w/o. Self-Explored method in Figure 5).

We fully agree that incorporating more baselines can further strengthen the robustness of our comparative analysis. Therefore, we have extended our experiments to include:

3. **Retrieval Augmentation**, which employs both sparse and dense retrieval to identify relevant documents based on target task instructions.
4. **Self-Instruct**, which utilizes the policy model to generate new instruction-response pairs for the target task.

| Methods                 | Qwen-2-7B-Instruct  |                   | Gemma-2-9b-Instruct |                   |
|-------------------------|---------------------|-------------------|---------------------|-------------------|
|                         | **In-House Test Tasks** | **Public Benchmarks** | **In-House Test Tasks** | **Public Benchmarks** |
| Prompting               |         36.7        |        27.2       |         67.0        |        33.7       |
| Rule-based QA           |         81.9        |        32.0       |         79.7        |        36.0       |
| Retrieval Augmentation  |         24.2        |        26.8       |         46.4        |        32.6       |
| Self-Instruct           |         55.8        |        31.6       |         76.9        |        35.1       |
| ADS                     |         **84.3**        |        **32.8**       |         **82.6**        |        **38.3**       |

**Compared to all the baseline methods, our ADS with iterative reinforcement learning processes have demonstrated superior performance improvements across both in-house test tasks and public benchmarks**, and maintaining its simplification without human intervention.  These results will be included in the revised manuscript for clarity and completeness.

> **Paper Revison**

We have carefully addressed all the comments and suggestions through comprehensive revisions. Below, we outline the major changes made to the manuscript (with key revisions highlighted in blue text in the PDF).

**Key Revisions**

1. We have elaborated on the reason for using in-context learning for policy model updating in **Section 4.4 (line 302-304)**. (Reviewer **CBK9, fuzK**)
2. We have included the comparison between more baseline methods and individual APIs in **Appendix E (line 1014-1025 Table 8)**, demonstrating that compared to all the baseline methods, the proposed ADS with all the APIs achieves improved performance across both in-house test tasks and public benchmarks. (Reviewer **CBK9, EL9t, 6jxd, eS6a, fuzK**)
3. We have discussed the influence of the Magpie instruction-following dataset in **Appendix F (line 1035-1046 Table 9)**, confirming that the improvements observed in our experiments are not attributed to the Magpie dataset. (Reviewer **CBK9**)
4. We have detailed the reasons for performance improvements achieved by the cost-control mechanism in **Section 6.2 (line 519-520)**. (Reviewer **eS6a**)
5. We have added the details to maintain distinctiveness between the newly generated instructions and the original instructions in **Appendix B (line 842-845)**. (Reviewer **eS6a**)
6. We have incorporated the analysis of tasks that ADS does not perform as expected in **Section 5.2 (line 453-455)**. (Reviewer **6jxd**)
7. We have fixed the typos and the hard-to-read sentences. (Reviewer **fuzK**)

We are deeply grateful for your insightful feedback, which has been instrumental in strengthening our work. We hope these revisions and additional analyses thoroughly address all raised concerns.

---

### Author Response · Authors · 2024-12-02

Dear Reviewers,

We sincerely appreciate your valuable feedback and the time you've dedicated to reviewing our paper. As we approach the final day of the discussion period, we kindly request any additional comments you may have on our revisions. We have invested significant effort in conducting additional experiments and addressing your queries, and would be grateful for your acknowledgment of our responses. Your feedback is crucial for us to effectively present our work to the research community. Please let us know if any points require further clarification.

Thank you very much and look forward to your replies!

Best regards,

Paper 6455 Authors

---

### Note · Authors · 2025-01-23

I have read and agree with the venue's withdrawal policy on behalf of myself and my co-authors.